# Integrative multi-omics analysis of intestinal organoid differentiation

Rik GH Lindeboom[1,†], Lisa van Voorthuijsen[1,†], Koen C Oost[2], Maria J Rodríguez-Colman[2], Maria V Luna-Velez[1], Cristina Furlan[1], Floriane Baraille[3], Pascal WTC Jansen[1], Agnès Ribeiro[3], Boudewijn MT Burgering[2], Hugo J Snippert[2,*] & Michiel Vermeulen[1,**] iD

## Abstract

Intestinal organoids accurately recapitulate epithelial homeostasis *in vivo*, thereby representing a powerful *in vitro* system to investigate lineage specification and cellular differentiation. Here, we applied a multi-omics framework on stem cell-enriched and stem cell-depleted mouse intestinal organoids to obtain a holistic view of the molecular mechanisms that drive differential gene expression during adult intestinal stem cell differentiation. Our data revealed a global rewiring of the transcriptome and proteome between intestinal stem cells and enterocytes, with the majority of dynamic protein expression being transcription-driven. Integrating absolute mRNA and protein copy numbers revealed post-transcriptional regulation of gene expression. Probing the epigenetic landscape identified a large number of cell-type-specific regulatory elements, which revealed Hnf4g as a major driver of enterocyte differentiation. In summary, by applying an integrative systems biology approach, we uncovered multiple layers of gene expression regulation, which contribute to lineage specification and plasticity of the mouse small intestinal epithelium.

**Keywords** adult intestinal stem cells; enterocytes; Hnf4g; organoids; systems biology

**Subject Categories** Development & Differentiation; Genome-Scale & Integrative Biology; Stem Cells

**Mol Syst Biol. (2018) 14: e8227**

## Introduction

The mammalian small intestinal epithelium is organized in glandular structures called crypts of Lieberkühn and finger-like structures called villi. New cells are constantly generated in the crypts to replace differentiated cells at the flanks of the villi, making the intestinal epithelium one of the fastest proliferating mammalian tissues. The driving force behind this constant self-renewal are adult stem cells that are located at the crypt base and divide once a day. These stem cells, marked by expression of the leucine-rich repeat G-protein-coupled receptor Lgr5 (Barker *et al*, 2007), give rise to a dividing pool of progenitor cells that migrate upward to differentiate and populate the villi with various specialized cell types that are involved in processes such as nutrient uptake, secretion of hormones, and mucus secretion. Interestingly, the intestinal epithelium displays remarkable cellular plasticity. Models making use of irradiation and specific depletion of the stem cell pool have demonstrated rapid regeneration of fully functional crypt units, suggesting that plasticity of non-proliferative cells contributes to the regeneration process (Tian *et al*, 2011; Metcalfe *et al*, 2014). Indeed, recent evidence suggests that a variety of committed progenitor cells can convert back to Lgr5[+] stem cells upon external stimuli to ensure intestinal homeostasis (van Es *et al*, 2012; Tetteh *et al*, 2016; Jadhav *et al*, 2017; Yan *et al*, 2017). The molecular mechanisms, which facilitate this cellular plasticity, are just becoming to be unraveled. Among others, previous work has revealed a broadly permissive chromatin structure that remains present upon differentiation of Lgr5[+] stem cells toward secretory and enterocyte precursors, suggesting that modular expression of one or a few transcription factors may be sufficient to trigger a global rewiring of gene expression and cellular phenotype. Indeed, broadly permissive chromatin that remains shared between cell-type progenitors provides a possible molecular explanation for the high degree of cellular plasticity in the intestinal epithelium (Kim *et al*, 2014).

Studying gene expression regulation in adult intestinal stem cells and their differentiated progeny has been quite challenging. Mouse models only provide limited amounts of material for resource-demanding technologies, while protocols to culture these cells *in vitro* have been lacking. However, recently, it was shown that isolated Lgr5[+] adult stem cells from the mouse gut can form "miniguts" or organoids *in vitro* in a well-defined semi-solid culture medium supplemented with essential growth factors (Sato *et al*,

1 Department of Molecular Biology, Faculty of Science, Radboud Institute for Molecular Life Sciences, Oncode Institute, Radboud University Nijmegen, Nijmegen, The Netherlands
2 Molecular Cancer Research, Center for Molecular Medicine, Oncode Institute, University Medical Center Utrecht, University Utrecht, Utrecht, The Netherlands
3 Centre de Recherche des Cordeliers, INSERM, IHU ICAN, Sorbonne Université, Université Paris Descartes, Paris, France
*Corresponding author. Tel: +31 887568989; E-mail: h.j.g.snippert@umcutrecht.nl
**Corresponding author. Tel: +31 243610562; E-mail: michiel.vermeulen@science.ru.nl
†These authors contributed equally to this work

2009). These organoids self-organize into an epithelial structure that is phenotypically and functionally reminiscent of the *in vivo* situation in the intestine of an adult mouse. Most importantly, all major intestinal cell types are continuously generated from the proliferative stem cell/progenitor pool followed by differentiation and migration toward their correct positioning along the crypt–villus axis. Intestinal organoids have great potential to study cell-type specification in the intestine, but their inherent multicellular heterogeneity represents a major drawback when combined with systems biology approaches to study gene expression regulation per cellular lineage.

Here, we show that minor modifications of the organoid culture medium to generate cell-type enriched mouse intestinal organoids, e.g., stem cell or enterocyte, can be used in combination with a multi-omics framework to decipher the molecular mechanisms that drive cell fate changes in small intestinal organoids. Our data reveal a global rewiring of the proteome and transcriptome during adult intestinal stem cell differentiation, with the large majority of dynamic protein expression being accompanied by a change in the corresponding transcript. Furthermore, we show that the Polycomb machinery represses a small but important subset of transcripts during adult intestinal stem cell differentiation. Finally, we identified the nuclear receptor hepatocyte nuclear factor 4 gamma (Hnf4g) as a major driver of enterocyte-specific gene expression patterns in intestinal organoids. Remarkably, loss of Hnf4g results in a partial loss of enterocyte-specific gene expression and an increase in secretory cells such as goblet cells in organoids and in the mouse small intestine. These datasets provide a rich resource for the community. Moreover, being applicable to all types of organoid cultures, including of human origin, our workflows represent a blueprint for future endeavors aimed at deciphering gene expression regulation in heterogeneous epithelial organoid cultures.

## Results

### Generation of stem cell-enriched and stem cell-depleted mouse small intestinal organoid cultures

To study *in vitro* differentiation of adult intestinal stem cells in a controlled manner, we generated cell-type enriched mouse small intestinal organoid cultures. To generate Lgr5$^+$ intestinal stem cell-enriched organoids, we supplemented the organoid culture medium (ENR; EGF, Noggin, R-Spondin; Sato *et al*, 2009) with CHIR99021 and valproic acid (CV), to influence Wnt and Notch signaling, respectively (Yin *et al*, 2014; Kishida *et al*, 2017). To induce overall differentiation, mainly toward enterocytes, we removed R-Spondin, a Wnt signaling enhancer, from the culture medium (EN) to mimic the diminishing Wnt gradient along the crypt flanks (Fig 1A; Farin *et al*, 2016; Yin *et al*, 2014). As a reference, we compared cell-type-enriched organoids to organoids grown in regular culture medium (ENR). To evaluate the efficiency of stem cell enrichment and depletion in CV and EN organoids, respectively, we made use of organoids containing a GFP reporter at the endogenous Lgr5 locus, thus enabling visualization of the adult Lgr5$^+$ intestinal stem cells (Tian *et al*, 2011; Fig 1B). Reassuringly, while Lgr5$^+$ stem cells are restricted to the bottom of the crypts in ENR organoids, they become the dominating cell type in CV organoids, while they are absent after R-Spondin removal in EN organoids. To evaluate whether *in vitro*

differentiation of organoids recapitulates *in vivo* differentiation along the intestinal crypt–villus axis, we compared deep proteomes of the organoid cultures (> 8,600 identified proteins) to proteomes from freshly isolated crypts and villi from the mouse small intestine (Fig 1C). When comparing dynamically expressed proteins in CV, ENR, and EN organoids to crypt and villi samples, we observed a striking correlation between the EN and the villus proteome, while ENR and CV show a reduced correlation. Thus, in agreement with inactivity of the Wnt pathway to stimulate enterocyte differentiation, the molecular signature of the EN organoid culture closely resembles the *in vivo* villus proteome, which is made up by more than 80% of enterocytes (van der Flier & Clevers, 2009). To determine whether these cell-type enrichment methods are mouse strain independent, we compared the proteome of CV, ENR, and EN cultured small intestinal organoids from different genetic backgrounds (Fig EV1C). Reassuringly, a strong correlation between the enriched organoids cultures is observed. This indicates that cell-type-enriched organoid cultures can be used as an *in vitro* culture system to study adult intestinal stem cell maintenance and differentiation in a controlled manner. In this study, we employed a multi-modal framework of state-of-the-art techniques to study the global molecular landscape of these organoid cultures in an unbiased and comprehensive manner (Fig 1D).

### Global rewiring of the proteome and transcriptome during adult intestinal stem cell differentiation

To study gene expression dynamics in the different organoid cultures at the mRNA and protein level, we integrated mass spectrometry-based proteomics and RNA sequencing data. We made use of spike-ins, both at the mRNA and at protein level, thus facilitating absolute quantification of the proteome and transcriptome. Reassuringly, both at the protein and at mRNA level, intestinal stem cell markers are significantly enriched and depleted in CV and EN, respectively, while known differentiation markers show opposite dynamics (Figs 2A and EV1A, Dataset EV1). As expected, both stem cell and differentiation markers are expressed in the ENR organoid culture. Interestingly, while the EN organoids are strongly enriched for enterocyte markers Vil1 (Sato *et al*, 2011), Alpi (Tetteh *et al*, 2016), Ptk6 (Haegebarth *et al*, 2006), Krt20 and Anpep (Merlos-Suarez *et al*, 2011), other differentiation markers are not enriched in EN compared to ENR. These include Defa6, Muc2, Dclk1, and Reg4, which are markers for Paneth, goblet, tuft, and enteroendocrine cells, respectively. This further indicates that the EN organoid culture is strongly enriched for mature enterocytes.

Given the fact that, both *in vivo* and *in vitro*, turnover of intestinal cells can be as short as 2 days, we hypothesized that time constraints only allow a minimal differential gene expression program to facilitate phenotypic changes between stem cells and mature enterocytes. Strikingly, however, we observed a global rewiring of the proteome during differentiation (Fig 2A). Of the 7,459 proteins that could be accurately quantified on both the mRNA and protein level, 59% are significantly changing in abundance in our differentiation model. Furthermore, the large majority of dynamic protein expression is being accompanied by changes at the transcriptome level, indicating that the observed rewiring of gene expression has a transcriptional foundation. As reported before (Schwanhausser *et al*, 2011), absolute mRNA and protein copy

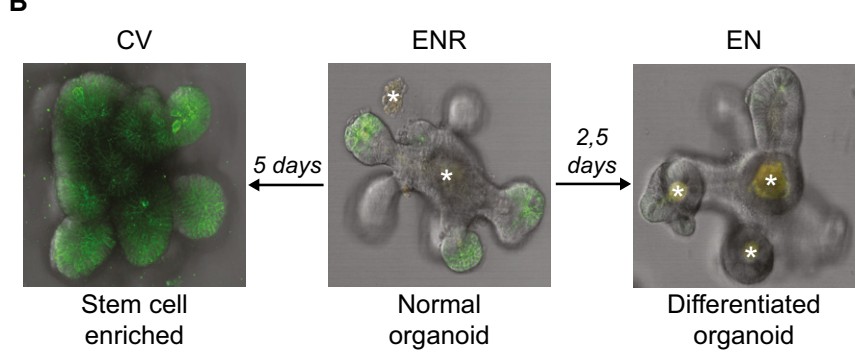

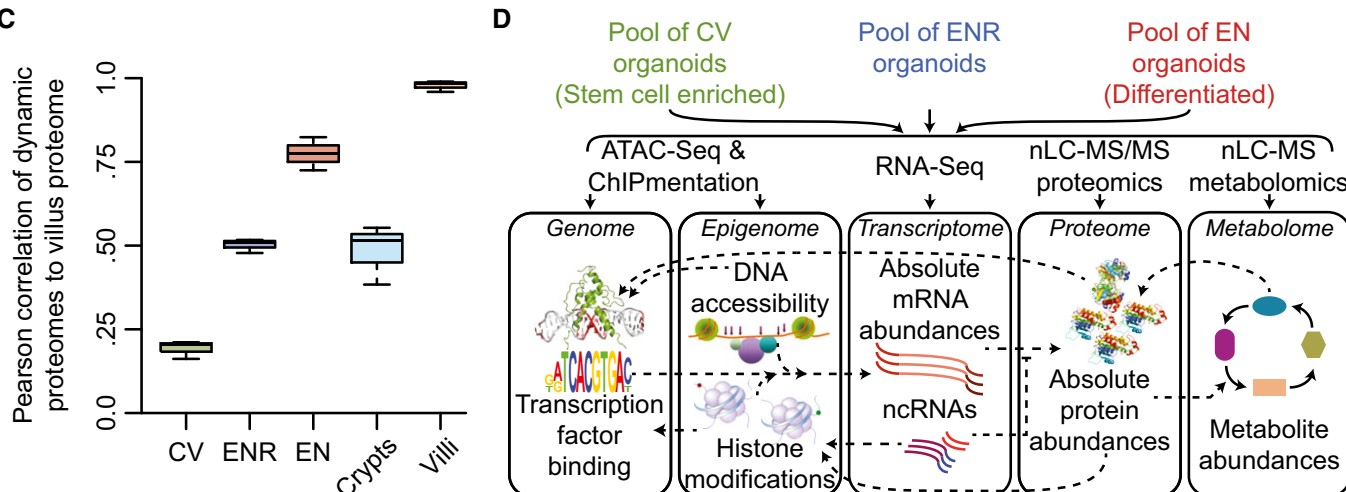

**Figure 1. Small-molecule-driven cell-type enrichment as a model for intestinal differentiation.**

A  Illustration of the small intestinal crypt–villus structure and the organoid culture conditions that are used to enrich for specific cell types.

B  Fluorescence microscopy images of typical mouse small intestinal organoids under different culture conditions. Lgr5+ intestinal stem cells are labeled with GFP, which are ubiquitously present, restricted to the bottom of the crypts and absent in CV, ENR and EN, respectively. * is auto-fluorescence.

C  Pearson correlations between villus proteome and the different organoid samples when focusing on proteins that are significantly changing between CV, ENR, and EN organoids, where the EN proteome correlates best with the villus proteome. The central line in each boxplot represents the median, the hinges are the first and third quartile, and the whiskers extend to the lowest and highest values within 1.5× the interquartile range from the hinges. Each sample was measured three times.

D  Schematic overview of the multi-omics approach that was taken to get a system-wide profile of cell-type-enriched organoid cultures.

numbers correlate poorly when compared within the same sample ($R^2 = 0.40$, Fig EV2A). However, here we show that transcriptomic changes between intestinal cell fates lead to correlated changes at the proteome (Fig 2A). As expected, the stem cell-enriched gene expression signature is significantly enriched for proteins involved in cell cycle and DNA replication, while enterocyte-enriched organoids express many proteins involved in different metabolic pathways (Fig 2B). This likely reflects the nutrient absorption function of mature enterocytes *in vivo*. To investigate the consequences of these abundant metabolic programs, we performed mass spectrometry-based lipidomics and metabolomics on the cell-type-enriched organoids and integrated this with gene expression dynamics (Fig EV1B, Dataset EV2). This revealed a correlation between the abundance of certain enzymatic activities and their involved metabolites. For example, we observed an upregulation

of enzymes related to glutathione metabolism coinciding with a strong upregulation of glutathione in the EN organoids. Interestingly, we observed a global decrease in glycerolipids in EN versus CV organoids, which could be linked to increased fat metabolism in EN organoids and the need for a steady supply of cellular organelles such as mitochondria in rapidly dividing stem cells. Indeed, it was recently shown that intestinal stem cells use mitochondria as their main source of energy (Rodriguez-Colman *et al*, 2017). Compared to ENR organoids, we observe that both mitochondrial proteins and mitochondrial DNA copies are higher in the stem cell-enriched population (CV) and in a lesser extent in the enterocyte-enriched condition (EN; Fig EV1E and F). In order to analyze the contribution of mitochondria to energy production, we performed bioenergetics analysis using Seahorse technology (Seahorse Bioscience). Interestingly, a hypermetabolic phenotype is

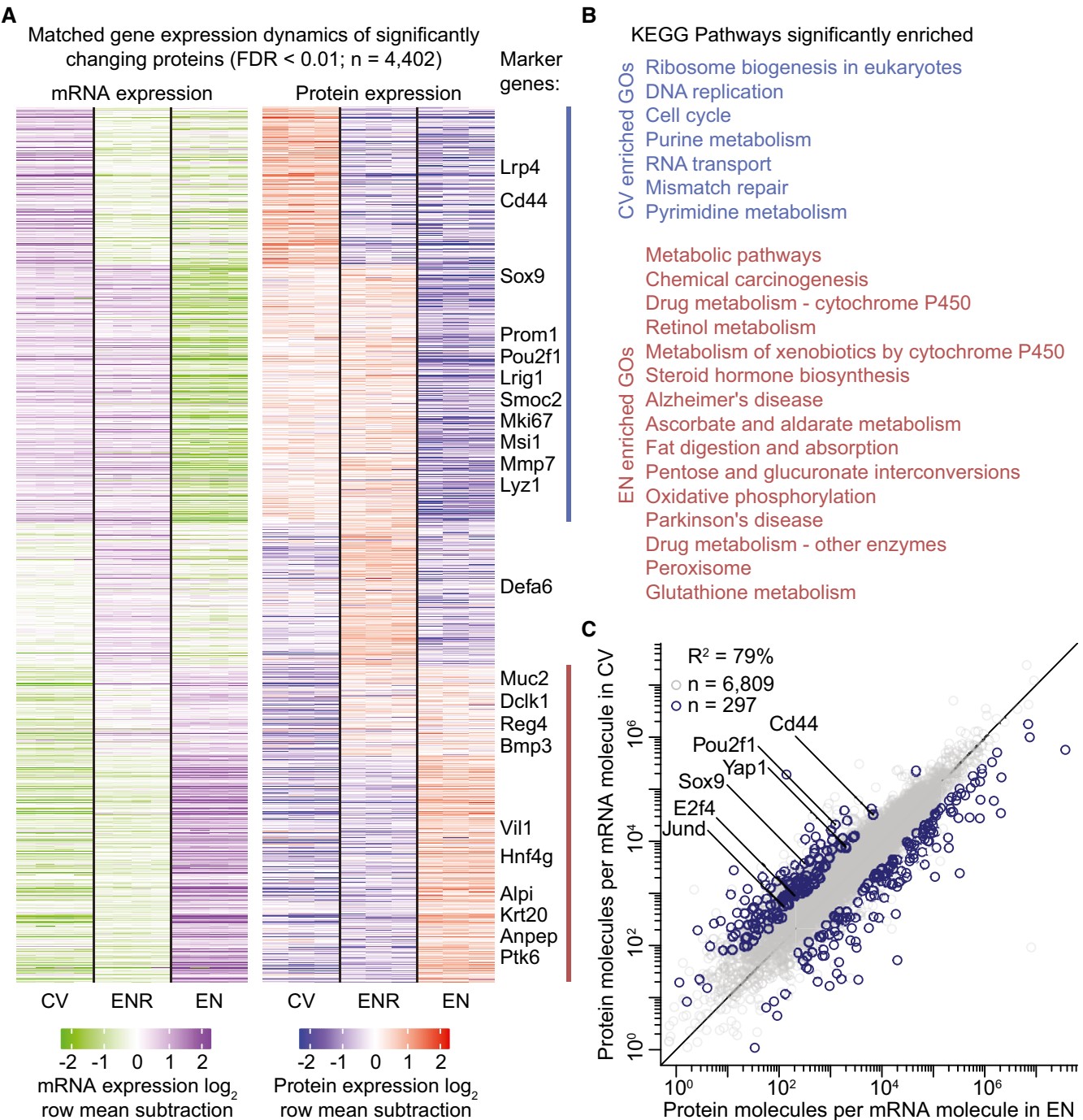

**Figure 2. Intestinal differentiation is coupled with global rewiring of gene expression.**

A   Two row-matched heatmaps showing the relative change in mRNA expression (left) and protein expression (right) of significantly changing proteins. Known markers of intestinal homeostasis are highlighted. Relative expression is shown as $\log_2$ fold change over the row mean.

B   KEGG pathways that are significantly enriched (FDR < 0.01) in stem cell-enriched organoids (blue) and differentiated organoids (red). Colors correspond to lines in (A) to show which gene clusters were used for enrichment analysis.

C   Scatterplot showing the amount of protein copy numbers per mRNA molecule in CV and EN organoids. Proteins with significantly changing protein over mRNA ratios (FDR < 0.05 and fold change > 3 in both CV/EN and ENR/EN) are plotted in blue. Significantly changing known markers of intestinal homeostasis are highlighted.

observed in the EN organoids based on increased glycolysis and mitochondrial respiration (Fig EV1D), while the relative contribution of mitochondrial respiration to cellular energy is higher in CV

organoids (Fig EV1G). Furthermore, we observed a stem cell-specific upregulation of several amino acids, which, together with the upregulated expression of the ribosome biogenesis pathway

could hint toward dynamic protein translation rates during adult intestinal stem cell differentiation.

Because amino acid levels can regulate translation efficiency of mRNAs through mTOR signaling (Zoncu *et al*, 2011), we investigated to what extent this could contribute to the observed proteome dynamics in the organoid cultures. Comparing protein copy numbers to mRNA copy numbers allows deducing the amount of proteins that are present per mRNA molecule, which acts as a measure for translation efficiency and protein stability. First, we verified that there are significant differences in protein translation and degradation rates between stem cell-enriched and enterocyte-enriched organoids (Fig EV2B). Next, we compared the protein to mRNA levels individually between CV and EN organoids (Fig 2C, Dataset EV3). Importantly, we see many known regulatory proteins of stem cell maintenance with significantly higher protein to mRNA ratios in the CV organoids. This could suggest that post-transcriptional regulation of transcription factors such as E2f4 and Sox9 might contribute to adult stem cell fate.

## Widespread epigenetic modulation facilitates transcription factor driven differentiation

Based on the observed extensive transcriptional regulation during enterocyte differentiation, we hypothesized that global remodeling of the epigenome must precede or at least accompany rewiring of the transcriptome and proteome. To investigate this, we used ATAC-seq to profile DNA accessibility and we performed ChIPmentation to map histone modifications H3K27ac, H3K4me3, and H3K27me3 in the organoid cultures. A total of 94,340 peaks could be detected in at least one replicate of the different sequencing datasets, which we used to identify genomic regulatory elements that undergo a significant change in DNA accessibility or histone modification occupancy between the different cell-type-enriched organoid cultures (Fig 3A, Dataset EV4).

Hyperaccessible regions, and regions marked with H3K27ac and H3K4me3, are associated with active regulatory elements in the genome such as enhancers and promoters. As expected, we found that a large portion of the epigenome is subjected to significant regulation of these activity marks; we observed significant dynamics in more than 12,000 identified genomic regions. We observed regulation of histone modification occupancies at both promoter and enhancer sites, while the most significant changes in DNA accessibility predominantly occur at TSS distant sites (enhancers). Reassuringly, changes in ATAC, H3K27ac, and H3K4me3 signal often correlate with each other, which further substantiates regulation by activation or repression of the identified regulatory elements. In contrast, a relatively small cluster of regulatory elements gain the transcriptionally repressive H3K27me3 Polycomb mark, which strongly anti-correlates with the active marks. This cluster contains regulatory loci associated with for example Lgr5 and Sox9 that are only active and open in stem cells and become repressed during differentiation. These observations are in agreement with previous studies showing that Polycomb repressive complex 2 (PRC2), which is responsible for deposition of H3K27me3, is essential for maintenance of the intestinal epithelium and intestinal homeostasis (Koppens *et al*, 2016). Furthermore, we see that the regulation of the epigenome is not restricted to isolated promoters and enhancers, but also happens at the level of much larger topologically associated

domains (TADs; Materials and Methods, Fig EV3C). Some of the TADs that show significant regulation contain important regulators of intestinal stem cells such as Sox9, Myc, and Ascl2, indicating that intestinal differentiation requires nuclear reorganization that leads to switching between so-called A and B compartments (Lieberman-Aiden *et al*, 2009) that contain regulators of intestinal homeostasis. To investigate to what extent the observed epigenetic changes contribute to transcriptional regulation in organoids, we employed a single-nearest-gene association rule to assign gene expression to identified regulatory elements. Indeed, we observe that gene expression dynamics globally correlates with dynamics of the active chromatin marks and DNA accessibility and anti-correlates with the heterochromatin mark (Fig 3B, left heatmap).

Since the observed rewiring of gene expression is transcriptionally regulated and associated with epigenetic remodeling, we set out to investigate which transcription factors could facilitate these observations. To this end, we used motif-score regression modeling on the differences in DNA accessibility and histone modification profiles. First, we reduced a list of non-redundant transcription factor motifs by filtering on transcription factor protein expression, and by LASSO and random forest feature selection (Materials and Methods). Next, we asked how important the selected motifs were to predict the dynamics in DNA accessibility and histone modification profiles of enterocyte-enriched organoids (Fig 3C). Remarkably, only a single transcription factor motif, namely Hnf4g, was selected that correlated with DNA accessibility and activity during differentiation. Hepatocyte nuclear factor 4 gamma (Hnf4g) is reported to be expressed in the small intestine (Bookout *et al*, 2006) and is known to be involved in intestinal development (Li *et al*, 2009), but its contribution as a global driver of epigenetic modulation and induction of gene expression during enterocyte differentiation was thus far unknown (Fig 3B, middle heatmap). In contrast, several transcription factor motifs were identified to correlate with active regulatory elements in ENR organoids (Fig 3C), which is likely a reflection of cellular heterogeneity in these ENR organoids. Motifs enriched in ENR organoids include Foxa2, a known regulator of enteroendocrine cell formation, and Nfkb1, which is reported to be important for intestinal homeostasis (Wullaert *et al*, 2011), and Sp1, which is known to be essential for intestine-specific gene expression through its interaction with Cdx2 (Shimakura *et al*, 2006).

To investigate whether the identified transcription factors are more abundant in the associated cell-type-enriched organoids, we looked at the mRNA and protein expression with respect to the ENR organoids (Fig 3D). Hnf4g shows a conspicuous high expression in the enterocyte-enriched organoids, while most of the other selected transcription factors are enriched in ENR organoids. Indeed, our absolutely quantified expression data show that enterocytes have a steady-state protein expression of more than a million Hnf4g molecules in every cell, clearly saturating every possible transcription factor binding site it can bind to. Interestingly, zinc finger and BTB domain-containing protein 7a and 7b (Zbtb7a and Zbtb7b), and ligand-dependent nuclear receptor corepressor (Lcor) are known transcriptional repressors that are more abundant in EN organoids, and their motifs correlate with regulatory elements that are repressed in these organoids. It is conceivable that these transcription factors contribute to enterocyte differentiation through epigenetic repression instead of classical activation.

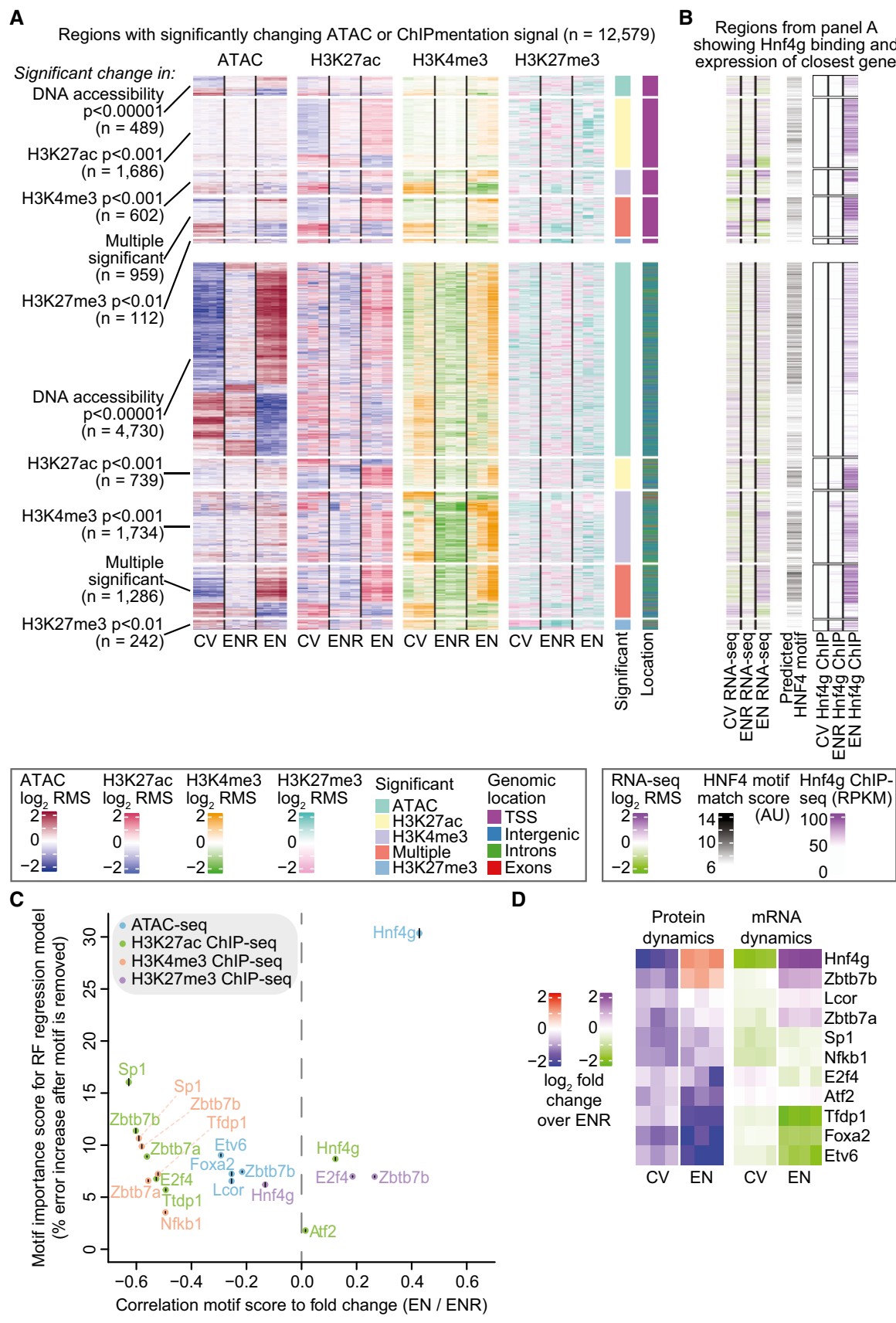

**Figure 3.**

**Figure 3.  Widespread modulation of epigenetic landscape for differentiated organoids is driven by Hnf4g.**

A    Heatmaps showing epigenome dynamics on peaks with significant changes in DNA accessibility and histone modifications between CV, ENR, and EN organoids. Heatmaps are split on their location near a transcription start site (< 500 bp from TSS, top heatmaps) or other location. Heatmaps are further subdivided based on the dataset it was found significantly changing in (shown at the left). Shown *P*-values are corrected for multiple testing hypotheses with the BH method.

B    Clustered genomic regions from (A) are shown. Regions were associated with their single closest TSS and corresponding mRNA expression changes are shown in the most left heatmaps. The best match score for the HNF4G motif is shown in the middle heatmap. Hnf4g binding is shown in the most right heatmaps.

C    Scatterplot with the transcription factors that are most likely to explain the dynamics shown in (A). The *x*-axis is the linear correlation between the best motif score and the $\log_2$-fold change EN over ENR under each significantly changing peak for a given dataset as shown in (A). The *y*-axis is the feature importance of the motif in a random forest model in percentage increase in mean-squared error (MSE), where motifs that are important for the model to accurately predict the given fold changes get a high error increase. Different datasets are color-coded. For motifs with more than one transcription factors associated to it, we show the transcription factor name with the most significant change in protein expression. The vertical line over each dot represents two times the standard deviation of the MSE.

D    Heatmaps showing change in protein (left) and mRNA (right) expression of selected transcription factors, compared to ENR organoids.

Data information: In panels (A and B), relative changes between different organoids are shown as $\log_2$ row mean subtractions (RMS) where the signal is compared to the row mean of all samples.

Given the strong association between Hnf4g binding and the observed epigenetic remodeling, we were interested to see whether the observed rewiring of gene expression could also be explained by Hnf4g binding. To this end, we focused on genes with significant upregulation in mRNA and protein expression in EN organoids to see which transcription factor binding sites were overrepresented in their promoters. Strikingly, we identify a Hnf4g binding motif in 62% of the upregulated genes (Fig EV3E), making it the most enriched and most abundant motif in these genes. The genes that can be activated by Hnf4g are significantly associated with metabolic pathways and protein digestion and absorption pathways, which represent the classical biological functions of enterocytes.

Finally, to validate the transcription factor model in which binding of Hnf4g can explain the epigenetic dynamics we observe during differentiation, we performed ChIPmentation for Hnf4g to identify the Hnf4g binding sites in CV, ENR, and EN organoids. Hnf4g binding is near absent in CV organoids whereas more than 12,000 Hnf4g peaks are identified in EN organoids (Fig 3B, right heatmap; Fig EV3D). Importantly, Hnf4g bound genomic regions in EN overlap with the predicted HNF4 motif sites, the open and active chromatin regions, and the associated gene expression in EN, whereas regulatory regions that are active in CV are devoid of Hnf4g binding. As an example, the promoter of the enterocyte marker Alpi is bound by Hnf4g in EN together with an increase in Alpi gene expression (Fig EV3B). Together, this indicates that Hnf4g is a driver of major epigenetic, transcriptomic and proteomic rewiring during enterocyte differentiation.

## Hnf4g drives enterocyte differentiation

Given the strong potential of Hnf4g to induce an enterocyte-specific epigenome and transcriptome in wild-type intestinal organoids, we set out to investigate the effect of Hnf4g loss in the small intestine using Hnf4g knockout (KO) mice and Hnf4g KO small intestinal organoids (Baraille *et al*, 2015; Fig EV3A). Transcriptome analysis of Hnf4g KO organoids reveals a significant loss of the enterocyte gene expression signature upon Hnf4g KO, while gene sets for secretory cells become significantly more abundant (Fig 4A). Previously, it was shown that loss of Hnf4g in mice results in increased numbers of enteroendocrine cells in the mouse intestine (Baraille *et al*, 2015). Here, we show in addition that upon a loss of Hnf4g increased numbers of goblet cells are observed *in vitro* and *in vivo* (Fig 4B and C). Moreover, goblet cell numbers show a higher degree of variation between individual villi (Fig 4E). Based on the observed

decrease in enterocyte-specific gene expression concomitant with an increase in differentiated secretory cells in Hnf4g-deficient mice and organoids, we conclude that Hnf4g is a major driver of enterocyte differentiation and that Hnf4g is essential to maintain a proper balance between differentiating cells from the secretory and absorptive lineage.

To investigate the molecular mechanisms that regulate Hnf4g-mediated differentiation, we examined the list of significantly changing genes with Hnf4g binding for potential regulators. Here, we found that hepatocyte nuclear factor alfa opposite strand (Hnf4aos) is the most significantly regulated long non-coding RNA in organoids and has a Hnf4g binding site with dynamic H3K4me3 occupancy at its promoter (Figs 4D and EV4C). Importantly, this enterocyte-specific lncRNA exhibits strong antisense transcription over one of the promoters of hepatocyte nuclear factor 4 alpha (Hnf4a). Hnf4a and Hnf4g are paralogs, but unlike Hnf4g, Hnf4a is ubiquitously expressed in both the intestinal crypt and villus (Sauvaget *et al*, 2002). Hnf4a is reported to be important for intestinal epithelial homeostasis by modulating Wnt signaling (Cattin *et al*, 2009) and can act as a tumor suppressor in colorectal cancer (Saandi *et al*, 2013). Interestingly, while the canonical Hnf4a transcript is not dynamically expressed in our system, we see a strong downregulation of an alternative isoform of Hnf4a during enterocyte differentiation. Here, we report that the antisense transcript Hnf4aos, presumably a direct target of Hnf4g, strongly anti-correlates with the expression of the alternative Hnf4a isoform it overlaps with, possibly through direct repression (Fig 4D).

To investigate the relevance of our findings for human biology and disease, we analyzed cancer stem cell (CSC) transcriptomes from patient-derived and CRISPR-engineered colorectal cancer organoids (Oost *et al*, 2018) for evidence of regulation by Hnf4g. Strikingly, Hnf4g is one of the most downregulated transcription factors in these CSCs (Fig EV4A). Consequently, the binding motif of Hnf4g is the most significantly enriched transcription factor motif in the promoters of genes that are specifically upregulated in those tumor cells that lack a CSC phenotype (Fig EV4B). This suggests that Hnf4g is playing a role in gene regulation even in transformed cancer cells, which is in line with recent reports on other cancer types, which revealed that dysregulated Hnf4g is associated with pancreatic (Klein *et al*, 2018), prostate (Shukla *et al*, 2017), and lung cancer (Wang *et al*, 2018). Overall, these observations illustrate the importance of system-wide interrogation and the power of integrative analyses to uncover novel biology related to intestinal homeostasis.

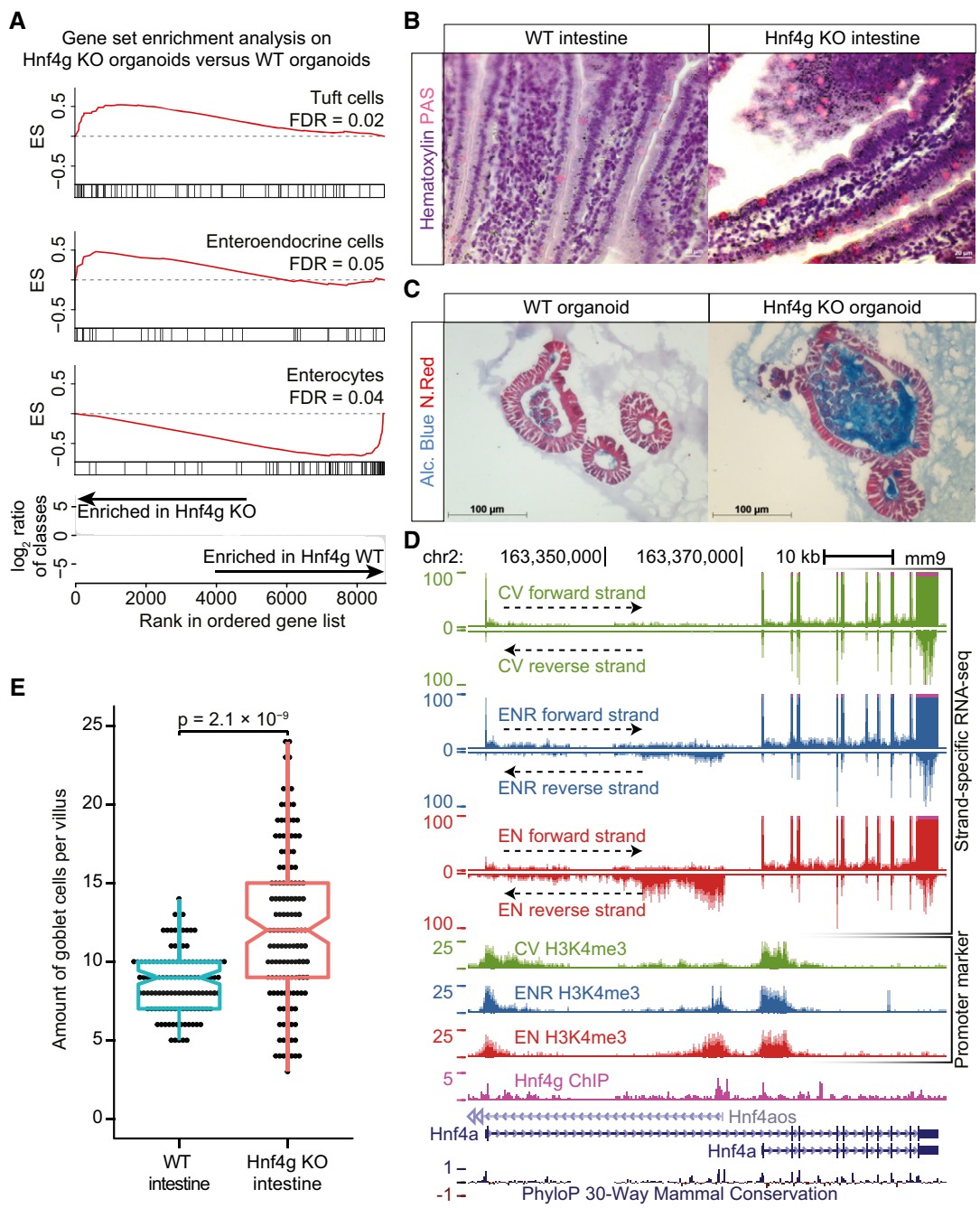

**Figure 4. Hnf4g drives differentiation of enterocytes.**

A   Gene Set Enrichment Analysis (GSEA) of Hnf4g KO organoids compared to WT organoids. Significantly changing intestinal cell-type gene sets from Haber *et al* (2017) are shown (FDR < 0.05).

B   Hematoxylin and PAS (periodic acid–Schiff) staining of WT (left) and Hnf4g KO (right) intestine. Nuclei are visualized using hematoxylin and goblet cells stain positive for PAS.

C   Alcian Blue and Nuclear Fast Red staining of WT (left) and Hnf4g KO (right) small intestinal organoids. Cells are visualized using Nuclear Fast Red. Intra- and extracellular mucus that is produced in goblet cells stain positive for Alcian blue.

D   Strand-specific RNA-seq data, Hnf4g ChIP-seq in EN, and promoter-specific histone modification H3K4me3 over the Hnf4a locus are shown. Data from different organoid cultures are color-coded. RNA-seq reads mapping to the positive strand (sense strand) are plotted in the top window of every sample, while RNA-seq reads that map to the negative strand are mirrored and shown in the bottom window of every sample. The two isoforms of Hnf4a that are expressed are shown in the bottom, together with the antisense transcript Hnf4aos. The "phyloP30way" track from the UCSC genome browser is plotted at the bottom to show sequence conservation.

E   Quantification of the amount of goblet cells per villus for WT and Hnf4g KO intestine. Dots represent the amount of goblet cells counted per villus. (*n* = 7–25 villi from 11 WT mice and 3–17 villi from 14 Hnf4g KO mice). *P*-value is from two-tailed Mann–Whitney *U*-test. The central line in each boxplot represents the median, the notch around this line is the approximate 95% confidence interval, the hinges are the first and third quartile, and the whiskers extend to the lowest and highest values within 1.5× the interquartile range from the hinges.

## Discussion

Here, we presented an integrative systems biology approach that in combination with organoid technology can disentangle the complex regulatory networks that underlie lineage specification in the mouse small intestine. Our study shows that profiling the epigenome, transcriptome, proteome, and metabolome represents a powerful approach to obtain a system-wide overview of the regulatory mechanisms, which enable gene expression regulation in cell-type-enriched organoid cultures. We show that, while each single technique is informative on its own, performing multiple complementary techniques will strengthen each other such that completely new questions can be answered after integrating a system-wide dataset.

The two major bottlenecks for employing a large set of "omics" techniques to study epithelial homeostasis in the intestine are (i) the need for large quantities of cellular material for high-resolution data generation. Especially to study rare cell types such as stem cells, this requires extensive animal resources. (ii) It is difficult to extract epithelial intrinsic processes from *in vivo* material due to variable interactions and signals from, for example the fluctuating microbiome, feeding patterns, and the immune system. To tackle these problems, intestinal organoids cultures were recently developed. Intestinal organoids can be cultured in fairly large amounts in defined medium while maintaining intestinal homeostasis that includes maintenance of an adult stem cell pool while also permitting differentiation to all major intestinal cell lineages. Through minor modifications of the culture medium, we have subsequently enriched for one of these cell types to study their characteristics in a comprehensive manner. Alternative methods that have been recently used to study differences between intestinal cell types include isolating specific cell types with fluorescence-activated cell sorting (FACS; Jadhav *et al*, 2017; Yan *et al*, 2017) and profiling intestinal samples at single-cell resolution (Haber *et al*, 2017). While such approaches have better discriminative power between cell types compared to our cell-type enrichment strategy, they are limited to techniques that are compatible with very small amounts of cells which often restricts the depth of the generated data and makes employment of several omics approaches in an integrative manner impossible. Moreover, in contrast to relying on mouse data, the platform as presented today is readily applicable to organoid cultures of human tissues, both from normal and from diseased origin.

While the focus of this study resulted in the discovery of a transcription factor as a regulator of intestinal stem cell differentiation, different analyses can be performed on our datasets as a resource to uncover additional layers of regulation, for example, when focusing on cell-type-specific metabolic processes or chromatin remodeling complexes. Indeed, we show that changes in the lipidome during differentiation reveal a change in mitochondrial load and energy metabolism. However, future efforts are needed to investigate the importance of other metabolic changes we observed, such as the drastic increase in glutathione, an important endogenous antioxidant capable of regulating redox signaling (Circu & Aw, 2012), and the global decrease in amino acids, which can affect post-transcriptional regulation (Zoncu *et al*, 2011). While we show that there could be possible post-transcriptional regulation of regulators of stem cell maintenance by integrating absolute gene expression data, additional experiments are required to further elucidate the importance of post-transcriptional regulation in the intestinal epithelium.

Finally, previous studies revealed that PRC2 is essential for stem cell maintenance (Koppens *et al*, 2016) and can directly regulate differentiation by repressing Atoh1 (Chiacchiera *et al*, 2016), an essential transcription factor for secretory cell differentiation. In line with this, our study also reveals regulation of Polycomb domains, and activation of TADs that harbor stem cell maintenance regulators. Together, this poses opportunities for future efforts to investigate how chromatin remodeling is changing the epigenetic landscape and what the interplay with the identified transcription factors is.

Enterocytes are replenished every 4–5 days, which requires an extremely fast cell fate switch from adult stem cell to functional absorptive enterocyte. Consequently, others have characterized Lgr5$^+$ stem cells as broadly permissive cell states at the epigenome level, to allow for quick transitions to differentiated cell types (Kim *et al*, 2014; Jadhav *et al*, 2017). Reanalyzing epigenetic data of FACS-sorted secretory and enterocyte progenitor cells shows that EN organoids are indeed differentiating toward the enterocyte lineage (Fig EV5A). Importantly, when comparing the histone marks and DNA accessibility of these progenitor cells, we detect a significant enrichment of the Hnf4g binding motif (Fig EV5B), validating its specificity to the enterocyte lineage. Using single-cell RNA sequencing, others have observed that the trajectory from intestinal stem cell to enterocyte is the shortest path in intestinal differentiation space (Yan *et al*, 2017). In contrast, we show that despite the short timeframe of differentiation, major changes in both gene expression and epigenetics are required for enterocyte differentiation. However, a large part of these changes can be attributed to a relatively simple differentiation trajectory, where a single transcription factor can be associated with most of the epigenetic dynamics, resulting in regulation of RNA expression which is directly translated in a proteomic landscape needed for mature enterocyte function.

Our integrative approach revealed that Hnf4g drives intestinal stem cell differentiation toward enterocytes. However, it is quite likely that other transcription factors regulate a smaller but equally important subset of genes that are essential to acquire an enterocyte phenotype. Indeed, very recently Regev and colleagues used single-cell RNA-seq to identify a handful of transcription factors that are specifically expressed in mature enterocytes, including Hnf4g (Haber *et al*, 2017). Interestingly, the same paper reports Hnf4a to be specific for intestinal stem cells, whose expression we found to be regulated by an Hnf4g-inducible non-coding RNA, postulating a possible self-reinforcing transcription factor switch between these two paralogs during adult intestinal stem cell commitment toward the enterocyte lineage.

Reassuringly, previously applied bioinformatics approaches also identified the HNF4 DNA binding motif to be most overrepresented in genes that are upregulated in enterocytes, but proposed Hnf4a as the central regulator for enterocyte differentiation (Stegmann *et al*, 2006). However, based on our absolute protein quantification and experimentally validated importance of Hnf4g, and in agreement with results from others (Sauvaget *et al*, 2002; Haber *et al*, 2017), we now show that Hnf4g is the most likely regulator of these genes. While we propose that Hnf4g is a major effector for enterocyte differentiation, Hnf4g knockout mice are viable with minor gastrointestinal complications (Baraille *et al*, 2015). While enterocytes are still present, intestinal differentiation is skewed toward the secretory cell lineage in Hnf4g$^{-/-}$ mice and Hnf4g$^{-/-}$ small intestinal organoids. This implicates that Hnf4g is indeed able to drive enterocyte

differentiation. However, we cannot exclude that other factors are likely to compensate for its absence from organogenesis onwards and can partially rescue a complete differentiation defect. Therefore, it will be of interest to study the molecular mechanisms that drive the residual enterocyte differentiation in Hnf4g$^{-/-}$ mice.

Organoid technology emerged as a powerful model system to study adult stem cell maintenance and differentiation for various organs, including liver, pancreas, and brain (Clevers, 2016). However, the inherent strength of these culture systems, namely cellular heterogeneity, also represents a challenge when combined with system-wide approaches to study gene expression regulation and cell fate switches per cellular lineage. Here, we have shown that small-molecule-driven perturbations to enrich for specific cell types represent a solution to this problem. Applying the system-wide approaches presented here to cell-type-enriched organoid cultures, complemented with time-course experiments and follow-up studies employing CRISPR-based perturbation and single-cell RNA-seq approaches (Dixit *et al*, 2016; Jaitin *et al*, 2016; Datlinger *et al*, 2017), represents a paradigm for future efforts aimed at deciphering gene expression regulation and cell fate switches in complex tissues.

# Materials and Methods

## Hnf4g knockout mouse model

Total and constitutive Hnf4g gene invalidation was performed as previously described (Baraille *et al*, 2015). Hnf4g$^{+/-}$ mice, obtained from Deltagen Company on a C57Bl/6J genetic background, were mated to obtain Hnf4g$^{-/-}$ mice on the same genetic background. In experiments, we compared Hnf4g$^{-/-}$ male mice with C57Bl/6J wild-type male Hnf4g$^{+/+}$ mice, matched in age and housed in the same room. All animals were housed in the SPF facility of the Centre de Recherche des Cordeliers on a 12 h-light/12 h-dark cycle and were fed with a standard diet (A03, Safe). Experimental procedures were in agreement with the French guidelines for animal studies from the Comité National de Réflexion Ethique sur l'Expérimentation Animale Charles Darwin (Ce5/2009/045).

## Organoid culture

Mouse small intestinal organoids Lgr5$^{GFPDTR/+}$ (Basak *et al*, 2017) were cultured as described previously (Sato *et al*, 2009). In short, the organoids were maintained using ENR medium which contained Dulbecco's modified Eagle's medium/Ham's F-12 supplemented with Pen/Strep, 10 mM HEPES, 1× Glutamax, 5% R-Spondin-1 (conditioned medium), 10% Noggin (conditioned medium), 1× B27 supplement, 1.25 mM *n*-acetyl cysteine, and recombinant mouse epidermal growth factor (50 ng/ml). Organoids were split every 4–5 days in a 1:4 ratio using mechanical dissociation and plated in fresh BME (Cultrex Reduced Growth Factor Membrane Extract, Type 2, PathClear). Stem cell-enriched organoid cultures (CV) were grown in ENR supplemented with CHIR99021 (3 μM) and valproic acid (1 mM) for 5 days. Stem cell-depleted organoid cultures (EN) were grown for 2.5 days using ENR medium without R-Spondin-1. Organoids were harvested by using mechanical dissociation followed by multiple ice-cold PBS washes (400 × *g*, 4°C, 5 min).

Organoids were aliquoted for proteomics, metabolomics, RNA sequencing, ATAC sequencing, and ChIPmentation.

## Crypt and villus sample preparation

The small intestine was harvested from a wild-type (C57BL/6), Lgr5$^{GFPDTR/+}$ (C57BL/6 and 129/ola), and Hnf4g$^{-/-}$ mouse and directly put on ice. We only used PBS without calcium and magnesium (PBS0) for harvesting of the crypts and villi. The small intestine was flushed three times with ice-cold PBS0 and cut open along its length. The villi of the small intestine were separated from the crypts using a cover glass, and the tissue was washed in ice-cold PBS0. Next, tissues were separately incubated for 30 min in 50-ml tubes with 25 ml ice-cold PBS0 with 5 mM EDTA. After incubation, 50-ml tubes were shaken five times. Now the supernatant contained crypts and villi, while the tissues were transferred to a new tube with 20 ml PBS0 to be shaken ten times again. The shaking and transferring step was repeated several times. Aliquots of the different supernatant samples were quickly examined under microscope for concentration and ratio between crypts and villi. Most optimal tubes were filtered with a 70-μm filter, and samples were collected by centrifugation at 800 rpm. For organoids culture, crypt pellets were embedded in BME and cultured in ENR supplemented for the first two passages with Y-27632 and Primocin. For the crypts and villi proteomic experiments, pellets were frozen prior to lysis. Frozen pellets were thawed and lysed in five volumes lysis buffer containing 1% NP-40, 0.5 M NaCl, 10% glycerol, 1× protease inhibitor cocktail (Roche), 0.5 mM DTT, and 0.1 M Tris–HCl at pH 8.5. Lysis was performed by vortexing at maximum speed for 1 min, 10 min on ice, and 1.5 h gentle rotation at 4°C. Finally, supernatant was directly used for proteomics mass spectrometry after centrifugation for 15 min at 21,100 × *g*.

## Organoid and BME proteomics sample preparation

SDS lysis buffer (4% SDS, 1 mM DTT, 100 mM Tris pH 7.5) was added to the organoid cell pellet and incubated for 3 min at 95°C. Samples were sonicated for five cycles (30 s on/30 s off, high) and spun down at 16,000 × *g* for 5 min. Supernatant was transferred to a new tube and stored at −80°C. UA buffer (8 M urea, 0.1 M Tris pH 8.5, 5 mM DTT) was added to the BME and incubated at room temperature for 30 min shaking at 800 rpm. Organoid and BME samples were used for FASP-SAX and FASP, respectively.

## Proteomics mass spectrometry

Protein lysates were digested with a trypsin/LysC combination using Filter Aided Sample-Preparation (FASP; Wisniewski *et al*, 2009b). For absolute quantification, 2 μg of a standard range of proteins (UPS2, Sigma) was added to each lysate (Schwanhausser *et al*, 2011). For deep-coverage proteomics, we subjected the digested peptides to a strong anion exchange (SAX; Wisniewski *et al*, 2009a) fractionation and collected a flow through, pH 11, pH 8, pH 5, and pH 2 fraction. Peptides were desalted and stored on StageTips (Rappsilber *et al*, 2007) prior to mass spectrometry analysis. Samples were applied to on-line Easy-nLC 1000 (Thermo) separation using different 214-min gradients of acetonitrile (7–30%, 5–23%, 8–27%, 9–30%, 11–32%, and 14–32% for unfractionated,

flow through, pH 11, 8, 5, and 2, respectively) followed by washes at 60% followed by 95% acetonitrile for 240 min of total data collection. Mass spectra were collected on a LTQ-Orbitrap Fusion Tribrid mass spectrometer (Thermo) in data-dependent top-speed mode with dynamic exclusion set at 60 s.

Raw mass spectra were analyzed in MaxQuant 1.5.1.0 (Cox & Mann, 2008) with match-between-runs, iBAQ, and label-free quantification enabled. A mouse RefSeq protein sequence database downloaded at 28-06-2016 from UCSC was used to identify proteins. Identified proteins were filtered for reverse hits and potential contaminants. Proteins detected in the BME gel were blacklisted for downstream analyses. We only considered proteins that were identified in at least all of the triplicates of the same sample, for downstream analyses. Missing values were semi-randomly imputed based on the assumption that they were not detected because they were under the detection limit of the mass spectrometer (in Perseus, MaxQuant software package). For absolute quantification, we applied a linear regression between supplied amounts and the iBAQ intensities of the spike-in proteins in the unfractionated sample, followed by a second linear regression between the absolute abundances and iBAQ intensities in the fractionated sample.

## Metabolomics sample preparation

For metabolomics triplicates of organoid samples, medium was removed from the well, washed with ice-cold 0.9% (w/v) NaCl, and transferred to 15-ml tubes. Tubes were centrifuged at 800 rpm for 3 min at 4°C, and the pellets were briefly trypsinized in order to remove BME (3 min at 37°C). Tubes were centrifuged at 800 rpm for 5 min at 4°C, and the organoid pellet was resuspended and homogenized in ice-cold methanol. Metabolite extraction was performed by mechanical disruption with silica beads. For Folch extraction, a volume of 200 µl sample in methanol is mixed with 100 µl internal standards in methanol and 75 µl of methanol. An amount of 750 µl dichloromethane is added and after vortex mixing the sample is incubated for 1 h at room temperature and regularly mixed. Phase separation is induced by adding 360 µl of ultrapure water. After vortex mixing the sample is allowed to stand at room temperature for 10 min after which it is centrifuged at $14,000 \times g$ for 10 min. The lower organic phase and upper aqueous phase are transferred to separate vials and evaporated to dryness. The residue of the organic phase is dissolved in 100 µl acetonitrile and subjected to lipidomics analysis. The residue of the aqueous phase is dissolved in 100 µl 10% methanol and subjected to metabolomics analysis.

## Lipidomics mass spectrometry

Lipidomics LC-MS/MS (data-dependent) analysis was performed using a Thermo Scientific Acella UHPLC system and an Acquity BEH C-18 column (1 × 100 mm, 1.7 µm). The column outlet was coupled to a Thermo Scientific Orbitrap XL equipped with an electrospray ion source operated in either negative or positive mode. The system was operated at a flow rate of 80 µl/min and 60°C. The mobile phases consisted of 40% acetonitrile in ultrapure water (v/v) also containing 10 mM ammonium acetate (solvent A), and 10% acetonitrile and 90% isopropanol (v/v) also containing 10 mM ammonium acetate (solvent B). A 10-min linear gradient of 40–100% B was started 1 min after the injection of the sample and

was at 100% B for the next 5 min. Thereafter, the system was returned to its starting situation. Total run time was 21 min. All acquired MS data were searched against available databases.

## Metabolomics mass spectrometry

Metabolomics LC-MS/MS (data-dependent) analysis was conducted with a Thermo Scientific Acella UHPLC system and an Acquity BEH C-8 column (1 × 150 mm, 1.7 µm). The column outlet was coupled to a Thermo Scientific Orbitrap XL equipped with an electrospray ion source operated in either negative or positive mode. The system was operated at a flow rate of 150 µl/min and 40°C. The mobile phases consisted of 6.5 mM ammonium carbonate pH 8 (solvent A), and 6.5 mM ammonium carbonate in 95% methanol (v/v, solvent B) in negative mode. For positive mode analysis, the solvents were 0.1% formic acid in ultrapure water and 0.1% formic acid in methanol, respectively. A 5-min linear gradient of 0–70% B was started 1 min after the injection of the sample. The gradient was increased to 98% B in 0.5 min and was at 98% B for the next 10 min. Thereafter, the system was returned to its starting situation. Total run time was 22 min. All acquired MS data were searched against available databases.

## mtDNA copy number

For DNA extraction, organoids were washed once with cold PBS and DNA was extracted with the QIAamp DNA Micro Kit (Qiagen). DNA was used as a template to amplify nuclear and mitochondria-encoded genes. qPCR was performed with FastStart SYBR Green Master mix (Roche) using the following primers:
B-Actin (CCCTACAGTGCTGTGGGTTT; GAGACATGCAAGGAGTGCAA),
DL (AATCTACCATCCTCCGTGAAACC; GCCCGGAGCGAGAAGAG),
ATP6 (AATTACAGGCTTCCGACACAAAC, TGGAATTAGTGAAATTGGAGTTCCT),
CytB (GCCACCTTGACCCGATTCT, TTCCTAGGGCCGCGATAAT),
ND2 (CACGATCAACTGAAGCAGCAA, ACGATGGCCAGGAGGATAATT).

## Bioenergetics

Seahorse Bioscience XFe24 Analyzer was used to measure extracellular acidification rates (ECAR) and oxygen consumption rates (OCR), and mitochondrial stress test was performed as previously described (Rodriguez-Colman et al, 2017). After Seahorse run, organoids were collected, DNA was extracted and quantified, and used for normalization.

## RNA sequencing sample preparation

RNA was extracted from snap-frozen cell pellets using the RNeasy RNA extraction kit (Qiagen) with DNaseI treatment. Ribosomal RNA depletion (Ribo-Zero Gold rRNA removal kit, Illumina) was performed using 725 ng of total RNA per sample followed by ethanol precipitation. 1 µl ERCC spike-in mix 1 (1:100, Ambion) was added to all samples during the rRNA depletion. rRNA-depleted samples were fragmented for 5 min at 95°C using 5× fragmentation buffer (200 mM Tris acetate, 500 mM potassium acetate, 150 mM magnesium acetate, pH 8.2) followed by ethanol precipitation. First-strand

cDNA synthesis was performed using Superscript III (Invitrogen) with the addition of random hexamer primers and actinomycin D (9 ng/ml). Afterward, second-strand cDNA synthesis (SuperScript, Invitrogen) was performed with the addition of dUNTPs and random hexamer primers. Samples were cleaned using Qiaquick MinElute Columns (Qiagen) after first- and second-strand cDNA synthesis. A maximum of 5 ng cDNA was used for sample preparation using the KAPA Hyper Prep Kit (KAPA Biosystems). NextFlex DNA barcodes (Bioo Scientific) were used for adapter ligation. Treatment for 15 min at 37°C with USER enzyme was done prior to library amplification. Libraries were amplified using 10 amplification cycles. Post-amplification cleanup was done using Qiaquick MinElute Columns (Qiagen) followed by size selection of 300 base pairs using E-Gel SizeSelect Agarose Gels 2% (Invitrogen). Library concentration was measured using the KAPA Library Quantification Kit (KAPA Biosystems), and library size was determined using the BioAnalyzer High Sensitivity DNA Kit (Agilent). Sequencing was performed using an Illumina HiSeq 2000, and 50-bp paired-end reads were generated.

### RNA sequencing analysis

Paired-end reads were aligned to mm9 using Genomic Short-read Alignment Program (Wu & Watanabe, 2005) with batch mode set to 5, novel splicing enabled and distance splice penalty set to 100. Transcript expression was quantified by kallisto (Bray *et al*, 2016) version 0.43.1 with a mouse RefSeq transcriptome index downloaded on June 28, 2016, from UCSC. One-hundred bootstrap samples were generated for each sample and strand-specific reads—first read reverse was enabled.

### Integration of proteome and transcriptome data

To accurately integrate the protein and transcript expression data, we used matched proteome and transcriptome RefSeq databases and quantified expression as described above. To ensure correct matching of isoforms, we restricted the transcriptome database to only transcripts that were already quantified at the protein level. Significantly regulated proteins were identified with an adapted Student's *t*-test on LFQ values while controlling for the multiple testing hypothesis with a 10,000× permutation-based false discovery rate [Perseus, MaxQuant software package (Cox & Mann, 2008)]. To calculate the amount of proteins per mRNA transcript, we performed absolute quantification of both mRNA and proteins as described above. Next, we calculated the protein over mRNA ratios by comparing all replicates to each other and taking the median ratio. *P*-values for differences in overall protein translation and degradation rates were determined by *t*-test on the Fisher-transformed Pearson correlations on protein over mRNA ratios. To test for significant differences in individual protein over mRNA ratios between organoid samples, we used a modified Student's *t*-test by comparing ratios from all replicates but by manually restricting the test to an appropriate amount of degrees of freedom.

### Gene ontology enrichment analysis

KEGG pathway enrichment analysis was done in DAVID 6.8 (Huang da *et al*, 2009). For enrichment analysis on proteomics data, a background protein list of all identified proteins was submitted. *P*-values were corrected by the Benjamini–Hochberg procedure.

Integrated gene expression and metabolomics Reactome pathway enrichment analysis were done in IMPaLA (Kamburov *et al*, 2011).

### ATAC sequencing

ATAC-seq was performed on approximately 50,000 cells. Organoids were resuspended in Recovery Cell Culture Freezing Medium (Gibco) and stored at −80°C prior to the ATAC-seq procedure. Once thawed, they were washed with PBS and ATAC-seq was performed as described in Buenrostro *et al* (2015) with three modifications. First, the total volume of the tagmentation reaction with in-house made Tn5 enzyme was halved. Second, the tagmentation reaction was stopped with 44 mM EDTA, 131 mM NaCl, 0.3% SDS, and 600 µg/ml proteinase K. Last, a reverse-phase 0.65× SPRI beads (Ampure) DNA purification was done after the first PCR. Amplified DNA was sequenced with an Illumina HiSeq 2000. Paired-end 50-bp sequencing reads were aligned to mm9 with BWA (Li & Durbin, 2009) allowing one mismatch. Reads were filtered for a quality score of at least 1, PCR duplicates were removed with Picard, and reads mapping to the mitochondrial chromosome were discarded. Peaks were identified with MACS 2.0 (Zhang *et al*, 2008) with a false discovery rate of 0.01.

### Chromatin harvesting

Cell pellets were dissolved in PBS and crosslinked in solution by using formaldehyde (1% final volume, shaking 10 min at room temperature). Glycine (0.125 M) was added to quench the reaction. Cell pellets were lysed in a volume of 110 µl using lysis buffer (20 mM HEPES pH 7.6, 1% SDS, 1× Protease Inhibitor Cocktails). Samples were sonicated using the Biorupter Pico (Diagenode) with six cycles (30 s on/30 s off). Afterward, the samples were spun at $16,200 \times g$ for 5 min at room temperature and the supernatant was stored at −80°C.

### ChIPmentation sample preparation

Chromatin from 80,000 to 150,000 cells was used for ChIPmentation. ChIPmentation was performed as described earlier (Schmidl *et al*, 2015) with several modifications. In short, the chromatin was incubated overnight at 4°C rotating in dilution buffer (1% Triton X-100, 1.2 mM EDTA, 16.7 mM Tris pH 8, 167 mM NaCl), 1× Protease Inhibitor Cocktail (Roche), and 1 µg of antibody [H3K4me3 (C15410003, Diagenode)/H3K27ac (C15410196, Diagenode)/H3K27me3 (C15410195, Diagenode), Hnf4g (Sigma, HPA005438)]. Per ChIP 10 µl protein A Dynabeads and 10 µl protein G Dynabeads were used. Beads were washed using dilution buffer (+0.15% SDS, +0.1% BSA) and incubated with the chromatin and antibody mix for 60 min at 4°C rotating. Afterward, the beads were washed at 4°C once with ChIP wash buffer 1 (2 mM EDTA, 20 mM Tris pH 8, 1% Triton X-100, 0.1% SDS, 150 mM NaCl), twice with ChIP wash buffer 2 (2 mM EDTA, 20 mM Tris pH 8, 1% Triton X-100, 0.1% SDS, 500 mM NaCl), and twice with TE (1 mM EDTA, 10 mM Tris pH 8). Beads were resuspended in Tagment DNA buffer and 1 µl Tn5 enzyme (produced in-house) and incubated for 10 min at 37°C with 550 rpm shaking. Afterward, the beads were washed twice with WBI (20 mM HEPES, 150 mM NaCl, 0.1% SDS, 0.1% DOC, 1% Triton X-100, 1 mM EDTA, 0.5 mM EGTA) and twice with WBIV (20 mM HEPES, 1 mM EDTA, 0.5 mM EGTA) with 5 min

rotating at room temperature in between. Samples were decrosslinked for 1 h at 55°C 1,000 rpm shaking followed by an overnight incubation at 65°C using elution buffer (0.5% SDS, 300 mM NaCl, 5 mM EDTA, 10 mM Tris pH 8) and proteinase K. Samples were incubated one additional hour the next day with elution buffer and proteinase K for 1 h at 55°C. The samples were purified using 2× SPRI AMPureXP beads. qPCR was used to determine the sufficient amount of PCR cycles needed to amplify the library. The libraries were amplified using the KAPA HiFi Hotstart Ready Mix (KAPA Biosystems) and Nextera Index Kit 1 (i7) and 2 (i5) primers (Illumina). Amplified libraries were purified using a 0.65× SPRI AMPureXP beads incubation followed by a 1.8× SPRI AMPureXP beads incubation. Library concentration was measured using the KAPA Library Quantification Kit (KAPA Biosystems); library size was determined using the BioAnalyzer High Sensitivity DNA Kit (Agilent). Sequencing was performed using an Illumina HiSeq 2000, and 50-bp paired-end reads were generated.

## ChIPmentation analysis

Paired-end sequencing reads were aligned to mm9 with BWA (Li & Durbin, 2009) allowing one mismatch. Reads were filtered for a quality score of at least 1, and PCR duplicates were removed with Picard. Peaks were identified with MACS 2.0 (Zhang *et al*, 2008) with a false discovery rate of 0.01.

## TAD activity analysis

To test for significant differences in enhancer and promoter activation at the topological associated domain level, we used publicly available TADs from Dixon *et al* (2012). We assigned every identified ATAC or ChIPmentation peak to a TAD and only considered TADs with at least 20 identified peaks for further analysis. Next, we quantified how much percent of the peaks had a fold change in H3K27ac signal in the same direction (percentage agreement). To identify a significance cutoff, we compared the results to 100 identical matrices with shuffled fold changes.

## Gene set enrichment analysis

Consensus cell-type-specific signature genes from Haber *et al* (2017) were used to find enrichment of intestinal cell types. The Lgr5$^+$ intestinal stem cell signature from Munoz *et al* (2012) was used as a stem cell gene set. We created a Lgr5$^-$ gene set from the data in Munoz *et al* (2012) in the same way the Lgr5$^+$ intestinal stem cell signature was defined. To calculate the false discovery rate of every GSEA, we performed 10,000 gene set perturbations.

## Transcription factor selection

To identify transcription factors that are most likely to explain observed dynamics in our DNA accessibility and ChIPmentation data, we used a non-redundant vertebrate motif database (https://doi.org/10.6084/m9.figshare.1555851) and selected motifs on transcription factor protein expression, and by LASSO and random forest feature selection. First, we reasoned that transcription factors that are not found in our deep proteome data are unlikely to contribute to gene expression. LASSO and random forest regression were

used to select the best candidate motifs to explain the observed epigenetic changes. LASSO reduces the number of features while maintaining high prediction accuracy. Random forest is an easily interpretable and robust machine-learning method that is able to identify non-linear relations and internally controls for overfitting. In more detail, to narrow down the list of possible transcription factor candidates, we first selected only motifs that are linked to transcription factors with significant dynamic protein expression in our data (FDR < 0.01). Next, we used the GimmeMotifs motif tool suite (van Heeringen & Veenstra, 2011) to calculate the motif score for all motifs under a union of all identified ATAC and ChIPmentation peaks. We then calculated noise-stabilized log$_2$-transformed fold changes in ATAC or ChIPmentation signal between ENR and EN by using DESeq2 (Love *et al*, 2014) to identify significantly changing regulatory elements. For the broad histone mark H3K27me3, we extended all peaks to 1,000 base pairs to ensure capturing robust signal. We used a tenfold cross-validated least absolute shrinkage and selection operator (LASSO; glmnet R package) to predict the significant log$_2$ fold changes with a matrix of motif scores and used its variable selection to further filter the possible motifs. Last, we used random forest regression (randomForest R package, 5,000 trees) on the remaining motif scores to predict the log$_2$ fold changes. To account for possible redundancy, we used the percentage increase in mean-squared error to select the motif with most predictive power, subtracted the variance that could be explained by that single motif, removed that motif from the matrix, and run the random forest regressor again on the residual variance. This was repeated until the individual motif that was selected as best predictor was not able to predict any residual variance on its own anymore. The whole analysis was repeated five times and only motifs that were always selected were kept. For every selected motif, the transcription factor with the most significant fold change in protein expression was shown as the most likely transcription factor candidate.

## Tissue Isolation and histology

Two- to six-month-old mice were euthanized, and jejuna were removed and flushed gently with phosphate-buffered saline (PBS). For histological analyses, pieces of proximal jejunum (1 cm) were immediately fixed overnight at 4°C in paraformaldehyde 4% before embedding in paraffin. Periodic acid–Schiff staining was performed by using a standard histological protocol (Cattin *et al*, 2009). Staining was examined by bright field microscopy using an Axiophot microscope connected to an Axiocam camera using the Axiovision 4.5 software (Zeiss). Goblet cells (PAS-positive cells) were counted in villi on one section of jejunum from WT and Hnf4g KO mice without mouse or tissue selection exclusion.

## Immunohistochemistry and staining organoids

Organoids in BME were fixed in 4% paraformaldehyde (PFA) for 15 min. To prevent depolymerization of BME hydrogel, 0.1% glutaraldehyde was added to the fixative. BME domes were incubated in 20% sucrose for 72 h at 4°C before being embedded in optimal cutting temperature (OCT) compound and snap-frozen. Cryotome sections of 4 μm thickness were subjected to hematoxylin and eosin staining for standard histology analysis. Acidic mucosubstances from

goblet cells were detected with Alcian blue solution (1 g of Alcian blue, pH 2.5, 3 ml/l of acetic acid, and 97 ml of distilled water). After 30-min incubation, Nuclear Fast Red (0.1 g of Nuclear Fast Red, 5 g of aluminum sulfate hydrate, and 100 ml of distilled water) was used to counterstain. Organoid section images were taken by bright field microscopy using an AxioCam MRc5 microscope (Zeiss).

### Live-cell confocal imaging

Live-cell images were captured with a Leica SP8X microscope. The organoids were held at 37°C in a microscope box and equipped with a culture chamber for humidity and 6.4% $CO_2$ overflow. Organoids were imaged in XYZ mode using a water 40× objective (HC PL APO CS2, N.A. 1.1). Post-acquisition analyses of phenotypes were performed manually using ImageJ.

### Western blot

For Hnf4g expression in WT and KO organoids: Organoid pellets were resuspended in 100 μl whole cell lysis buffer per well of a 6-well plate (150 mM NaCl, 50 mM Tris pH 8, 10% glycerol, 1% NP-40, 1 mM DTT, 1× CPI). Extracts were rotated for 1.5 h at 4°C followed by seven cycles of 30 s on and 30 s off sonication (Biorupter Pico, Diagenode). Afterward, the samples were spun at $16,200 \times g$ for 5 min at room temperature and the supernatant was stored at −80°C. Samples were loaded on an 8% SDS–PAGE gel and afterward transferred to a nitrocellulose membrane. Membranes were incubated with primary antibody in 5% milk/TBST overnight at 4°C [anti-beta actin 1:5,000 (Abcam, ab16039), and anti-HNF4G 1:500 (Sigma, HPA005438)]. Afterward, membranes were incubated with secondary antibody [Polyclonal Swine Anti-Rabbit Immunoglobulins/HRP 1:3,000 (Dako, P0399)] and imaged using chemiluminescent substrate (Thermo Fisher, 34580).

For mitochondrial ETC complexes: Organoids were washed once and Matrigel was mechanically removed in cold PBS. Total proteins were collected by direct lysis of organoids in Laemmli sample buffer. Proteins were run in SDS–PAGE and transferred to Polyscreen PVDF transfer membranes (PerkinElmer). Antibodies: Total OXPHOS Rodent WB Antibody Cocktail (ab110413, Abcam) and anti-vinculin (V9131, Sigma).

### Data availability

Next-generation sequencing data have been deposited at Gene Expression Omnibus (GEO) with accession code GSE114113. The mass spectrometry data have been deposited at the ProteomeXchange Consortium via the PRIDE partner repository with the dataset identifier PXD009700.

Expanded View for this article is available online.

### Acknowledgements

We thank all the members of the Vermeulen laboratory for fruitful discussions. We thank F. de Sauvage (Genentech) for providing organoid cultures of Lgr5-DTR-GFP knock-in mice, Deltagen Inc., for supplying the Hnf4g[+/−] mice, Eva Janssen-Megens from the Radboud University for help with next-generation sequencing, and Dr. Edwin Stigter from the Medical Metabolomics Facility UMCU for performing LC-MS/MS-based metabolomics. The Vermeulen, Snippert, and Burgering laboratories are part of the Oncode Institute which is partly financed by the Dutch Cancer Society (KWF). Furthermore, work in the Vermeulen and Burgering laboratory is funded by the gravitation program CancerGenomiCs.nl from the Netherlands Organization for Scientific Research (NWO).

### Author contributions

MV, HJS, RGHL, and LvV conceived the study. RGHL and LvV performed the proteomics, RNA-seq, ChIPmentation, and ATAC-seq experiments. LvV performed organoid culture. RGHL performed all analyses. KCO and HJS performed confocal microscopy experiments and isolated fresh crypt and villus samples. MJR-C and BMTB performed lipidomics and all metabolomics experiments. MVL-V performed all organoid stainings. FB and AR performed the Hnf4g KO mice experiments. CF and PWTCJ performed proteomics experiments. RGHL, LvV, and MV wrote the manuscript, which was reviewed by all authors.

### Conflict of interest

The authors declare that they have no conflict of interest.

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
