## [Review Process File · Molecular Systems Biology]

Integrative multi-omics analysis of intestinal organoid differentiation

Rik G.H. Lindeboom, Lisa van Voorthuisen, Koen C. Oost, Maria J. Rodríguez-Colman, Maria V. Luna-Velez, Cristina Furlan, Floriane Baraille, Pascal W.T.C. Jansen, Agnès Ribeiro, Boudewijn M.T. Burgering, Hugo J. Snippert and Michiel Vermeulen.

Review timeline:	Submission date:	16 th January 2018
	Editorial Decision:	12 th February 2018
	Revision received:	23 rd May 2018
	Editorial Decision:	23 rd May 2018
	Revision received:	1 st June 2018
	Accepted:	5 th June 2018

Editor: Maria Polychronidou

Transaction Report:

1st Editorial Decision

12th February 2018

Thank you again for submitting your work to Molecular Systems Biology. We have now heard back from the three referees who agreed to evaluate your study. As you will see below, the reviewers appreciate that the study seems potentially interesting and that the presented datasets might be useful for the field. They raise however a series of concerns, which we would ask you to address in a major revision.

Without repeating all the points listed below one of the more fundamental issues refers to the need to perform further analyses in order to better support the role of Hnf4g as a key regulator of enterocyte differentiation. Of course all other issues raised by the referees would need to be thoroughly addressed.

REFeree REPORTS

Reviewer #1:

In the manuscript entitled "System-wide rewiring of gene expression during intestinal organoid differentiation" perform extensive and thorough dissection of molecular mechanisms that drive differential gene expression during adult intestinal stem cell differentiation. They apply a number technologies, including RNA seq, genome wide accessibility studies using ATAC-seq and transcription factor binding approaches using ChIPmentation. Moreover they also apply proteomics approaches to provide an accurate understanding of a post-transcriptional regulatory network during enterocyte lineage specification. Finally the use of metabolomics and lipidomics enables a more

complete overview of the major changes at functional level. In addition, the authors also make use of publicly available datasets to extend the scope of their work, and to strengthen their hypotheses. Through this holistic approach, they reach the conclusion that during intestinal differentiation a global rewiring of the epigenome, transcriptome and the proteome takes place much faster than what was thought, and that the majority of the changes are driven by transcription. This process facilitated by the fact that the stem cells have their epigenome in a permissive state (Kim, 2014). They also propose that this global change can be, in fact, attributed to a single transcription factor, Hnf4g, that operates as the main driver of enterocyte specification. I found the work compelling. The datasets generated are important resources for the field. However, there are some parts that must be improved:

1. Authors do not present enough evidence to support the claim that Hnf4g is the key TF driving enterocyte differentiation. The authors predict Hnf4g binding motifs in differentiation genes, yet they should show Hnf4g binding to the regulatory regions of these genes in organoids under the distinct culturing conditions.
2. Likewise, overexpression experiments have been performed in cell lines rather than in organoids. It would also be important to include loss of Hnf4g function experiments (shRNA or CRISPR) in organoids to demonstrated altered differentiation.
3. The main driver of the stem cell program is WNT signaling. Is Hnf4g expression negatively regulated by WNT signaling? Does over expression of Hnf4g trigger downregulation of beta-catenin/TCF4 transcriptional activity?
4. In figure 2 authors show markers of intestinal homeostasis enriched in CV (stem cell genes) and in EN (enterocyte lineage specific genes). However, other well-known markers of enterocyte differentiation such as KRT20, ANPEP, CA1/2 are not shown. I would like to see if these genes are expressed in EN organoids. Moreover, authors could leverage their datasets to define novel markers of intestinal differentiation and link them to particular functions in enterocytes.

Minor points:

- Authors do not specify the time points needed to achieve CV nor EN enriched cultures. In Figure 1B, time must be indicated, as well as in all other experiments where CV or EN organoids were used.
- In figure 2 it is somewhat confusing where panel A ends, and panel B starts, panel B must be moved to the right, since there is space.
- In figure 3, panel A is too densely packed, should be broken down into simpler panels, easier to interpret. I cannot understand how the far most right column labeled HNF4g motif should be interpreted.

Reviewer #2:

The mammalian intestinal epithelium became a classical model system to study dynamics of chromatin states during adult stem cell maintenance and differentiation. Several studies mapped the distribution of histone marks associated with either transcriptional activity or repression in adult intestinal stem cells (ISCs) and their differentiated progenies, for example enterocytes, *in vivo*. Using genome-wide ChIP and ATAC-sequencing, potential cell type specific regulatory elements were identified. The distribution of histone modifications and chromatin accessibility were further associated with transcriptional activity of the nearest genes in both ISCs and their differentiated progenies. In summary, differentiation of the adult ISCs is accompanied by strong changes in the distribution of modified histones in genic and intergenic regions, which leads to transcriptional changes (thousands of genes get up- or downregulated). Moreover, the binding sites for a few transcription factors, including Hnf4a, Cdx2 and Gata4, were mapped by ChIP-sequencing in both ISCs and enterocytes. Functional study in mice showed that Hnf4a in concert with Cdx2 play an important role during ISC differentiation. Finally, the comparative proteomic analysis of ISCs and their differentiated progenies was also performed.

In the present study the authors used intestinal organoids to study transcriptional, chromatin,

proteome and metabolic changes accompanying ISC differentiation. The intestinal organoids were cultured in different conditions to either enrich or deplete for adult ISCs. The authors conclude that strong changes at the transcriptional and protein levels take place during ISC differentiation. Consistent with previous studies, they find that a few ISC signature genes acquire tri-methylation of K27 at histone H3 (H3K27me3) during differentiation. Using ATAC- and ChIP-sequencing they see a strong correlation between open chromatin sites and the presence of H3K27Ac and H3K4me3 histone marks. The genome-wide distribution of open chromatin sites as well as H3K27Ac strongly changes during differentiation. They further searched for potential transcription factor binding sites within dynamic regulatory elements and found Hnf4g motif. An overexpression of Hnf4g in LS174T cells results in up-regulation of a differentiation marker Alpi and downregulation of ISC and proliferation markers, Lgr5 and Mki67. The authors thus conclude that Hnf4g is a major driver of enterocyte specific transcriptional program. Finally, the authors performed profiling of metabolites in ISC enriched and depleted organoids. Yet, the functional studies were not performed to address whether the changes in metabolites have any importance.

The conclusions of the study are consistent with previously published literature. The study, however, does not bring significant new knowledge. The transcriptome and chromatin data (and conclusions) are already available from in vivo studies. The same is true for proteome. Although, in the present study 15% more proteins were identified compared to the study by Munoz et al., 2012, ISC markers Lgr5 and Ascl2 were not found by Lindeboom and colleagues.

Major points

1. To conclude whether Hnf4g is a major driver of enterocyte specific gene expression program, loss-of function study using CRISPR/Cas9 in organoids is required.
2. The metabolomics analysis table lacks P values. Therefore, it is not clear, which changes are meaningful.

Minor points:

The authors describe changes in metabolomics or H3K27me3, or TADs in the results part. Yet, it is not clear why. They do not refer to these data even in Discussion part.

Reviewer #3:

The manuscript by Lindeboom et al. describes an intestinal organoid culturing modification that generates most stem cell-enriched small intestinal organoids versus differentiated organoids enriched with enterocytes. Then, a system-wide "multi-omic" approach was used to full characterize the proteome, transcriptome, and epigenome of these different organoid states. From this systems level, analysis it is a little surprising that only Hnf4g transcription factor is identified to be a master regulator of differentiation. While this multi-omic dataset is potentially useful to the community, the biological insights derived from data modeling can be better well-supported by validation experiments.

Major points

1. Kunihiro et al., 2017, AJP. has already worked out the conditions for deriving enriched differentiated cell populations in organoids. Please compare and contrast the phenotypes obtained with the current ones.
2. It was never shown by the authors that the organoid culture derived from the EN condition is actually a pure enterocyte population. Since enterocytes still represent ~90% of the cells, you will get an enterocyte signature with bulk approaches. Immuno-staining of these EN organoids for canonical markers of other differentiate cell types will resolve this issue.
3. As above, better characterization of EN organoids. For instance, if these are differentiated, they would all be "differentiated out" and not be maintainable for more than a week after exposure to EN.
4. Likewise, without stem cell function, the organoids should not be able to be passaged, since there would be no renewal activity.
5. Some of the features hypothesized such as mitochondrial morphological and functional differences between CV and EN can be tested experimentally.
6. Stem cell organoids usually are represented by spherical cyst like structures due to lack of heterogeneous proliferation rates (for bud forming). Here, the stem cell organoids in CV still contains a lot of buds, which seems to contradict the canonical view such as the stem organoids

from the Rodriguez-Colman paper in 2017 (one of the co-authors here). Perhaps, the CV conditions only induces Lgr5 expression but not enriches stem cells. Please compare the gene expression profile of these organoids to that of Lgr5 flow-sorted cells in an analysis similar to 1C.

7. Validation of hypotheses. For instance, using a protein to mRNA ratio, the authors hypothesized that there could be translational / stability differences in important regulators. This is an area that can use some experimental validation such as ribosomal profiling/tagging, etc.

8. Similar to above, the authors predicted increased in a large number of binding sites for Hnf4g in EN organoids. This should be testable with an Hnf4g CHIP experiment.

9. A better validation experiment as opposed to the Hnf4g oe experiment in a cancer cell line is to demonstrate perturbation of Hnf4g by crispr/cas9 in organoids, specifically showing that Hnf4g KO organoids are impaired towards enterocyte differentiation in EN.

10. A missed opportunity is to compare epigenetic data in organoids to in vivo data generated by the Shivdasani lab (quoted by the author). Specifically, the comment on whether the open nature of chromatin in the intestinal system exists in organoids not clarified.

Minor points

1. I am not sure if the small fonts in many of the figures will translate well in print.

1st Revision - authors' response

23rd May 2018

12th Feb 2018

RE: MSB-18-8227 System-wide rewiring of gene expression during intestinal organoid differentiation

Reviewer #1:

In the manuscript entitled "System-wide rewiring of gene expression during intestinal organoid differentiation" perform extensive and thorough dissection of molecular mechanisms that drive differential gene expression during adult intestinal stem cell differentiation. They apply a number of technologies, including RNA seq, genome wide accessibility studies using ATAC-seq and transcription factor binding approaches using ChIPmentation. Moreover they also apply proteomics approaches to provide an accurate understanding of a post-transcriptional regulatory network during enterocyte lineage specification. Finally the use of metabolomics and lipidomics enables a more complete overview of the major changes at functional level. In addition, the authors also make use of publicly available datasets to extend the scope of their work, and to strengthen their hypotheses. Through this holistic approach, they reach the conclusion that during intestinal differentiation a global rewiring of the epigenome, transcriptome and the proteome takes place much faster than what was thought, and that the majority of the changes are driven by transcription. This process is facilitated by the fact that the stem cells have their epigenome in a permissive state (Kim, 2014). They also propose that this global change can be, in fact, attributed to a single transcription factor, Hnf4g, that operates as the main driver of enterocyte specification. I found the work compelling. The datasets generated are important resources for the field. However, there are some parts that must be improved:

We thank the reviewer for his or her positive and constructive feedback on our manuscript.

1. Authors do not present enough evidence to support the claim that Hnf4g is the key TF driving enterocyte differentiation. The authors predict Hnf4g binding motifs in differentiation genes, yet they should show Hnf4g binding to the regulatory regions of these genes in organoids under the distinct culturing conditions.

This is a valid point. To address this, we performed ChIP-sequencing for Hnf4g in the three different organoid cultures (CV / ENR / EN) (*Widespread epigenetic modulation facilitates transcription factor driven differentiation, page 6, paragraph 6*). In agreement with our proteomics and genomics data, we observe a near absence of Hnf4g binding in stem cell enriched organoids (87 binding sites), whereas many (12357) Hnf4g binding sites are observed in differentiated (EN) organoid cultures (Figure EV3D).

Importantly, the observed binding of Hnf4g in EN organoids overlaps with the HNF4 motif and the open and active chromatin regions in EN (Figure 3B).

2. Likewise, overexpression experiments have been performed in cell lines rather than in organoids. It would also be important to include loss of Hnf4g function experiments (shRNA or CRISPR) in organoids to demonstrate altered differentiation.

Again, this is a valid point. In the revised manuscript we replaced the HNF4g overexpression experiment in human colorectal cells with Hnf4g KO experiments in mice and organoids (*Hnf4g drives enterocyte differentiation, page 6, paragraph 1*). We obtained Hnf4g KO organoids by isolating small intestines from Hnf4g KO mice (Baraille et al. 2015, Diabetes). Using RNA sequencing, we observed an enrichment of expressed genes that are specific for secretory cells concomitant with a depletion of enterocyte-specific genes in these Hnf4g KO organoids, indicating that Hnf4g is important for driving expression of enterocyte specific genes (Figure 4A). In agreement with these observations, staining of goblet cells in the small intestinal epithelium of Hnf4g KO mice and Hnf4g KO organoids revealed a significant increase in goblet cell numbers compared to WT intestines and organoids (Figure 4B and 4C). Altogether, these experiments revealed disrupted intestinal epithelial homeostasis in the absence of Hnf4g, characterized by skewed differentiation towards secretory cells.

3. The main driver of the stem cell program is WNT signaling. Is Hnf4g expression negatively regulated by WNT signaling? Does over expression of Hnf4g trigger downregulation of beta-catenin/TCF4 transcriptional activity?

Intestinal stem cells located at the crypt base are exposed to a high concentration of Wnt, whereas cells located in the villus, are not. Our cell type enrichment in organoids mimics this Wnt gradient by supplementing the medium with CHIR99021 or by removing R-spondin to generate stem cell enriched or depleted organoids, respectively. In our model, we observe a strong upregulation of Hnf4g when the Wnt signals are removed, suggesting that Wnt signaling is regulating Hnf4g expression, at least indirectly. However, to investigate whether Hnf4g and Wnt signaling are directly regulating each other, we investigated the activation of WNT signaling pathway signature in the transcriptome of Hnf4g KO organoids. In these organoids we did not observe differential expression of Wnt responsive genes compared to WT organoids (Reviewer Figure 1), suggesting that Hnf4g is not a direct regulator of Wnt signaling. However, we believe that further research is needed to decipher the exact molecular interplay between Wnt signaling and enterocyte factors such as Hnf4g, but such experiments are in our opinion beyond the scope of the current manuscript.

Reviewer Figure 1: Gene Set Enrichment Analysis (GSEA) of Hnf4g KO organoids compared to WT organoids for Wnt responsive genes.

4. In figure 2 authors show markers of intestinal homeostasis enriched in CV (stem cell genes) and in EN (enterocyte lineage specific genes). However, other well-known markers of enterocyte differentiation such as KRT20, ANPEP, CA1/2 are not shown. I would like to see if these genes are expressed in EN organoids. Moreover, authors could leverage their datasets to define novel markers of intestinal differentiation and link them to particular functions in enterocytes.

We thank the reviewer for pointing this out. We mined our transcriptome and proteome data for expression of these marker genes (*Global rewiring of the proteome and transcriptome during adult intestinal stem cell differentiation, page 3, paragraph 1*). Krt20 and Anpep are significantly upregulated in EN organoids, both at the transcript and protein level. Both markers are now highlighted in Figure 2A. We did not detect Ca1/2 expression in our organoids cultures, both at the mRNA and protein level. Besides the identification of Hnf4g as a major driver of differentiation, we indeed detect many more genes and pathways that show specific expression in our cell type enriched organoids. As suggested by the reviewer, we now included a section in the discussion where we highlight the potential of this study as a resource to define enterocyte and stem cell specific gene sets (*Discussion, page 7, paragraph 3*).

Minor points:

- Authors do not specify the time points needed to achieve CV nor EN enriched cultures. In Figure 1B, time must be indicated, as well as in all other experiments where CV or EN organoids were used.

We thank the reviewer for pointing out the missing description of the CV and EN culturing method, this is indeed essential information. We added the culturing conditions and timing to the methods sections (*Organoid culture, page 9, paragraph 1*), and indicated the culturing time points in Figure 1B.

- In figure 2 it is somewhat confusing where panel A ends, and panel B starts, panel B must be moved to the right, since there is space.

We moved panel B from figure 2 to the right, hopefully this makes the separation of the different panels more clear.

- In figure 3, panel A is too densely packed, should be broken down into simpler panels, easier to interpret. I cannot understand how the far most right column labeled HNF4g motif should be interpreted.

In the revised manuscript we split Figure 3A into two separate panels. Furthermore, we moved the legends down to create more space between heatmaps and we added more descriptive labels. Panel A now only shows clustered dynamics in DNA accessibility and histone marks, while panel B now contains the expression dynamics of the closest gene, the HNF4 motif and the newly generated Hnf4g ChIP-seq data. The HNF4 motif and Hnf4g binding data reveals that clusters of accessible and active genomic regions in enterocytes display strong binding of Hnf4g, while the stem cell specific accessible regions are devoid of Hnf4g binding. We hope that these data in panel B are now easier to interpret and link to the observed epigenetic dynamics in panel A.

Reviewer #2:

The mammalian intestinal epithelium became a classical model system to study dynamics of chromatin states during adult stem cell maintenance and differentiation. Several studies mapped the distribution of histone marks associated with either transcriptional activity or repression in adult intestinal stem cells (ISCs) and their differentiated progenies, for example enterocytes, in vivo. Using genome-wide ChIP and ATAC-sequencing, potential cell type specific regulatory elements were identified. The distribution of histone modifications and chromatin accessibility were further associated with transcriptional activity of the nearest genes in both ISCs and their differentiated progenies. In summary, differentiation of the adult ISCs is accompanied by strong changes in the distribution of modified histones in genic and intergenic regions, which leads to transcriptional changes (thousands of genes get up- or downregulated). Moreover, the binding sites for a few transcription factors, including Hnf4a, Cdx2 and Gata4, were mapped by ChIP-sequencing in both ISCs and enterocytes. Functional study in mice showed that Hnf4a in concert with Cdx2 play an important role during ISC differentiation. Finally, the comparative proteomic analysis of ISCs and their differentiated progenies was also performed.

In the present study the authors used intestinal organoids to study transcriptional, chromatin, proteome and metabolic changes accompanying ISC differentiation. The intestinal organoids were cultured in different conditions to either enrich or deplete for adult ISCs. The authors conclude that strong changes at the transcriptional and protein levels take place during ISC differentiation. Consistent with previous studies, they find that a few ISC signature genes acquire tri-methylation of K27 at histone H3 (H3K27me3) during differentiation. Using ATAC- and ChIP-sequencing they see a strong correlation between open chromatin sites and the presence of H3K27Ac and H3K4me3 histone marks. The genome-wide distribution of open chromatin sites as well as H3K27Ac strongly changes during differentiation. They further searched for potential transcription factor binding sites within dynamic regulatory elements and found Hnf4g motif. An overexpression of Hnf4g in LS174T cells results in up-regulation of a differentiation marker Alpi and downregulation of ISC and proliferation markers, Lgr5 and Mki67. The authors thus conclude that Hnf4g is a major driver of enterocyte specific transcriptional program. Finally, the authors performed profiling of metabolites in ISC enriched and depleted organoids. Yet, the functional studies were not performed to address whether the changes in metabolites have any importance.

The conclusions of the study are consistent with previously published literature. The study, however, does not bring significant new knowledge. The transcriptome and chromatin data (and conclusions) are already available from in vivo studies. The same is true for proteome. Although, in the present study 15% more proteins were identified compared to the study by Munoz et al., 2012, ISC markers Lgr5 and Ascl2 were not found by Lindeboom and colleagues.

We thank the reviewer for his or her constructive feedback. Indeed, the intestinal epithelium is a well-studied model for adult stem cell maintenance and differentiation and many seminal studies have been published in recent years. In our current manuscript, we, for the first time, used a holistic approach to interrogate many aspects of intestinal stem cell differentiation in an integrative manner. We respectfully disagree with the point raised by the reviewer regarding the

lack of significant new knowledge brought by our study. While some of our findings indeed nicely agree with literature, our approach allowed us to answer fundamental questions related to intestinal stem cell differentiation. In addition to uncovering previously underappreciated global changes at the epigenome, transcriptome and proteome level during lineage commitment in the intestinal epithelium, we are for example, to the best of our knowledge, the first to report a transcription factor (Hnf4g) that specifically drives enterocyte differentiation. In our revised manuscript, we further strengthened this observation by performing experiments on Hnf4g KO mice and organoids (*Hnf4g drives enterocyte differentiation, page 6, paragraph 1*), and profiled the Hnf4g binding in organoid cultures using ChIP-seq (*Widespread epigenetic modulation facilitates transcription factor driven differentiation, page 6, paragraph 6*).

Moreover, moving beyond mouse models, the established systems biology platform that we present in this manuscript has the potential to be applicable to all types of organoid cultures, which include human organoid cultures of normal and diseased origin. To illustrate the relevance of our findings for human biology and disease, we analyzed human colorectal cancer stem cell (CSC) transcriptomes for evidence of regulation by Hnf4g (*Hnf4g drives enterocyte differentiation, page 6, paragraph 3*). Strikingly, Hnf4g is one of the most downregulated transcription factors in CSCs (Figure EV4A). Consequently, the binding motif of Hnf4g is the most significantly enriched transcription factor motif in the promoters of genes that are specifically upregulated in tumor progression organoid cells that lack a CSC phenotype (Figure EV4B). This suggests that Hnf4g is playing a role in gene expression regulation in cancer cells, which is in line with recent reports linking perturbed Hnf4g function to pancreatic (Klein et al., Nature Communications, 2018), prostate (Shukla et al, Cancer Cell, 2018) and lung cancer (Wang et al., Oncotarget, 2017).

Reviewer Figure 2: relative abundance Lgr5 and Ascl2 in the different culture conditions. Mass spectrometry intensities of Lgr5 and Ascl2 in three replicates per organoid condition are shown.

Finally, our proteomics dataset exceeds any currently available intestine proteome datasets in depth, and it does actually cover the stem cell markers (Ascl2 and Lgr5) mentioned by the reviewer. As seen in Reviewer Figure 2, both Ascl2 and Lgr5 display strong upregulation in stem cell enriched organoids. The reason that these proteins are not included in Figure 2 is that we filter for proteins which are identified with at least 2 unique peptides and three independent identifications by mass spec in at least three replicates to ensure high confidence quantification of protein abundance. Lgr5 and Ascl2 did not meet these criteria, likely because even in stem cells they have low expression (Lgr5) or because they are very small proteins with few tryptic peptides (Ascl2). Nevertheless, we increased the confidence and depth of proteome datasets by repeating the mass spectrometry experiments (*Generation of stem cell enriched and -depleted mouse small intestinal organoid cultures, page 3, paragraph 1*).

Importantly, we observe high correlations and reproducible dynamics between the cell type enriched organoid cultures (Figure EV1C).

Major points

1. To conclude whether Hnf4g is a major driver of enterocyte specific gene expression program, loss-of function study using CRISPR/Cas9 in organoids is required.

This is a valid point, see our response to reviewer 1, point 2.

2. The metabolomics analysis table lacks P values. Therefore, it is not clear, which changes are meaningful.

We thank the reviewer for pointing this out. We now visualized and included false discovery rates in both Figure EV1B and Dataset EV2.

Minor points:

The authors describe changes in metabolomics or H3K27me3, or TADs in the results part. Yet, it is not clear why. They do not refer to these data even in Discussion part.

We agree with the reviewer that these datasets are underused and not thoroughly discussed. In the revised manuscript we now included functional follow-up experiments based on observations from the lipidomics experiments (*Global rewiring of the proteome and transcriptome during adult intestinal stem cell differentiation, page 4, paragraph 2*). Based on the global upregulation of glycerophospholipids in our stem cell enriched organoids, we hypothesized that the rapidly dividing stem cells require more organelles such as mitochondria. We experimentally tested this hypothesis with live-cell metabolic assays (Seahorse technology, Bioscience), by quantifying metabolic complexes with western-blot, and by qPCR on mitochondrial DNA. Indeed, we found that stem cell enriched organoids have a higher mitochondria load, revealing cell type specific differences in metabolic activity and energy use in the intestine (Fig EV1D-G), thus illustrating the usefulness of our metabolomics data as a resource. In addition, we now included a discussion section about H3K27me3 domains and changing TAD activity (*Discussion, page 7, paragraph 3*).

Reviewer #3:

The manuscript by Lindeboom et al. describes an intestinal organoid culturing modification that generates most stem cell-enriched small intestinal organoids versus differentiated organoids enriched with enterocytes. Then, a system-wide "multi-omic" approach was used to full characterize the proteome, transcriptome, and epigenome of these different organoid states. From this systems level, analysis it is a little surprising that only Hnf4g transcription factor is identified to be a master regulator of differentiation. While this multi-omic dataset is potentially useful to the community, the biological insights derived from data modeling can be better well-supported by validation experiments.

We thank the reviewer for his or her constructive feedback on our manuscript.

Major points

1. Kunihiro et al., 2017, AJP. has already worked out the conditions for deriving enriched differentiated cell populations in organoids. Please compare and contrast the phenotypes obtained with the current ones.

Reviewer Figure 3: gene expression of enterocyte enrichment cultures in small intestinal organoids. Relative expression of cell type specific markers in enterocyte enriched cultures compared to ENR gene expression. Gene expression is normalized over Actb expression and fold changes are shown in log₂. Error bars show 1x the standard deviation.

We thank the reviewer for pointing out this paper which describes different culture conditions to achieve cell type enrichments in mouse small intestinal organoids. In the revised manuscript we cite this paper in the introduction section. We compared the enterocyte enrichment culture method (C59 + VA) of Kishida et al. (2017) with our enterocyte enrichment method (EN) (Reviewer Figure 3), since the stem cell enrichment method used in the paper and our manuscript are similar (CV). Both cultures were harvested after a differentiation period of 2,5 days. The enrichment methods both show a decrease of the stem cell marker Lgr5 and the Paneth cell marker Lyz1 compared to a typical ENR culture, which is also shown in Figure 1b of Kishida et al. (2017). As discussed in the manuscript, Hnf4a has an alternative transcript, which is downregulated during enterocyte differentiation. Both culture conditions show a reduction of this Hnf4a transcript. The enterocyte differentiation markers Hnf4g and Alpi are increased in both culture conditions, which confirms that both culture methods enrich for enterocytes. We conclude that both the EN and the C59 + VA method can be used to enrich for enterocytes in intestinal organoid cultures.

Since we do not focus on the development of cell type specific enrichment methods in our manuscript, this figure is not included in the expanded view of the revised manuscript, but we only provide this figure in our rebuttal for the reviewers.

2. It was never shown by the authors that the organoid culture derived from the EN condition is actually a pure enterocyte population. Since enterocytes still represent ~90% of the cells, you will get an enterocyte signature with bulk approaches. Immuno-staining of these EN organoids for canonical markers of other differentiated cell types will resolve this issue.

It is indeed correct that the EN culture does not consist of a pure enterocyte population. The EN culture is used to obtain enterocyte enriched organoids. As depicted in Figure 2A of the manuscript, we observe an upregulation of enterocyte markers like Vill, Ptk6 and Alpi in the EN condition compared to CV and ENR. However, other differentiated cell types are also present in the EN condition as shown by the expression of Muc2, Dclk1, and Reg4 (for goblet cells, tuft cells, and enteroendocrine cells respectively). Importantly, the expression of enterocyte markers is the strongest in EN organoids compared to the ENR condition, which indicates a strong enrichment of enterocytes compared to secretory cell types.

In addition to the observed dynamics in Lgr5-DTR-GFP+ cells, we set out to visualize differentiated cell types in the CV, ENR and EN organoids, as the reviewer suggested. While we were technically not able to visualize enterocyte cells by microscopy, we used Alcian Blue staining to visualize

goblet cells in CV, ENR and EN (see Reviewer Figure 4), which revealed an absence and enrichment for goblet cells in CV and EN organoids, respectively. However, as pointed out by the reviewer, enterocytes are by far the most dominant cell type in absolute abundance in the intestinal epithelium, which is why the enterocyte signature is dominant in the bulk assays that we performed.

Reviewer Figure 4: Alcian blue and Nuclear Fast Red staining of CV, ENR, and EN small intestinal organoids. Stainings are performed on fixed WT organoids in the different culture conditions. Alcian blue is used to visualize the goblet cells and the cells are visualized using nuclear fast red.

3. As above, better characterization of EN organoids. For instance, if these are differentiated, they would all be "differentiated out" and not be maintainable for more than a week after exposure to EN.

We indeed forgot to mention the duration of differentiation using the EN culture method, we now included this in our methods section (*Organoid culture, page 9, paragraph 1*) and in Figure 1B. It is indeed correct that the organoids stop proliferating and die after continued culturing on EN medium. We passage and expand organoids on ENR medium, followed by induction of differentiation by washing out R-spondin. We cultured these EN organoids for 2,5 days after which we harvested the organoids. Apologies for the confusion this may have caused.

4. Likewise, without stem cell function, the organoids should not be able to be passaged, since there would be no renewal activity. It is indeed also not possible to passage the EN organoids, as mentioned above, we now included a methods section for this and highlighted the timepoints used in Figure 1B.

5. Some of the features hypothesized such as mitochondrial morphological and functional differences between CV and EN can be tested experimentally.

We now tested the hypothesized differences in mitochondrial load between CV and EN (*Global rewiring of the proteome and transcriptome during adult intestinal stem cell differentiation, page 4, paragraph 2*). Western-blot and qPCR for mitochondria shows an increased presence of mitochondria in the CV organoids (Fig EV1E-F). Reassuringly, when investigating bioenergetics with Seahorse technology (Biosciences) we observe an increase in relative mitochondrial respiration metabolic activity in the CV population (Fig EV1D,G). These results are in agreement with the decrease in glycerolipids observed in the EN population and reveal differences in energy metabolism in different intestinal cell types.

6. Stem cell organoids usually are represented by spherical cyst like

structures due to lack of heterogeneous proliferation rates (for bud forming). Here, the stem cell organoids in CV still contains a lot of buds, which seems to contradict the canonical view such as the stem organoids from the Rodríguez-Colman paper in 2017 (one of the co-authors here). Perhaps, the CV conditions only induces Lgr5 expression but not enriches stem cells. Please compare the gene expression profile of these organoids to that of Lgr5 flow-sorted cells in an analysis similar to 1C.

Organoids growing in medium containing high Wnt3a concentrations (WENR) are indeed growing in cyst like structures due to homogeneous proliferation, as also shown by Rodríguez-Colman et al. (2017, Nature). These WENR organoids, which resemble embryonic small intestine organoids and have decreased Lgr5 expression, are used by Rodríguez-Colman et al. (2017, Nature) to study the differentiation process before bud formation. Crypt structures are a typical phenotype for CV organoids and have increased expression of Lgr5 (Kishida et al. (2017, AJPGLP), Yin et al. (2014, Nature Methods)). Together, this suggest that CV organoids are enriched for adult intestinal stem cells as shown by increased Lgr5 expression, WENR organoids have a decrease in Lgr5 expression because of their more embryonic state.

As suggested by the reviewer, we compared our gene expression data with Lgr5 flow-sorted cells from Muñoz et al. (2012, The EMBO journal) which revealed a strong enrichment of the Lgr5+ gene signature in the CV population (Figure EV1A) (*Global rewiring of the proteome and transcriptome during adult intestinal stem cell differentiation, page 3, paragraph 1*). This again validates the stem cell identity of the CV organoid population.

7. Validation of hypotheses. For instance, using a protein to mRNA ratio, the authors hypothesized that there could a translational / stability differences in important regulators. This is an area that can use some experimental validation such as ribosomal profiling/tagging, etc.

We agree that further research is needed to experimentally validate the changes in proteins / mRNA ratios. The interesting observation of potential post transcriptional regulation of certain transcripts could be of great interest to the field, and the method we use to identify post transcriptional regulated transcripts takes a unique angle to address this problem, which could be explored by others. However, we now emphasize in the revised manuscript that these observations are predictive only, and require further functional validation (*Discussion, page 7, paragraph 3*). Such validations are, however, beyond the scope of the current manuscript.

8. Similar to above, the authors predicted increased in a large number of binding sites for Hnf4g in EN organoids. This should be testable with an Hnf4g CHIP experiment.

To address this valid point, we performed Hnf4g ChIP-sequencing in the three organoid culture conditions (CV / ENR / EN) (*Widespread epigenetic modulation facilitates transcription factor driven differentiation, page 6, paragraph 6*). Strikingly, global Hnf4g binding is observed in EN (12357 binding sites) compared to ENR and CV organoids (1619 and 87 binding sites, respectively; Figure EV3D). Importantly, Hnf4g binding as observed by ChIP-seq overlaps with open and active chromatin regions in the EN culture, which shows that Hnf4g is indeed binding to these enterocyte specific regulatory elements (Figure 3B).

9. A better validation experiment as opposed to the Hnf4g oe experiment in a cancer cell line is to demonstrate perturbation of Hnf4g by crispr/cas9 in organoids, specifically showing that Hnf4g KO organoids are impaired towards enterocyte differentiation in EN.

This is a valid point, see our response to reviewer 1, point 2.

10. A missed opportunity is to compare epigenetic data in organoids to in vivo data generated by the Shivdasani lab (quoted by the author). Specifically, the comment on whether the open nature of chromatin in the intestinal system exists in organoids not clarified.

This is a very good suggestion. We integrated the mentioned datasets with the DNA accessibility and histone mark profiles we generated (*Discussion,*

page 8, paragraph 4). Reassuringly, we see that the enterocyte specific dynamics we observe is most similar to the profiles from enterocyte precursor sorted cells, while our stem cell enriched organoids show dynamics that are more similar to their Lgr5+ and secretory progenitor populations (Figure EV5A).

As already mentioned in the manuscript, our results partially challenge the conclusions made by the Shivdasani lab that the intestinal epithelium is characterized by a globally permissive epigenetic landscape. While we focus on terminally differentiated cells, their study focusses on progenitor cells that are likely in a more permissive state. On the other hand, it is conceivable that global and extensive changes are less pronounced due to technical differences. However, because of the technical and biological differences with our study, we do not wish to make bold statements about this, as it goes beyond the scope of our manuscript. Interestingly, when investigating predicted Hnf4g binding in secretory / enterocyte progenitors in the Shivdasani dataset, a significant enrichment of the Hnf4g motif is observed in regions that are more active / open in enterocyte progenitors compared to secretory progenitors (Figure EV5B) (Discussion, page 8, paragraph 4). This strengthens our observation that Hnf4g is an enterocyte specific factor, and a not pan-differentiation factor. Furthermore, this observation indicates that Hnf4g is already of importance during the progenitor stage of differentiation.

Minor points

1. I am not sure if the small fonts in many of the figures will translate well in print.

We thank the reviewer for pointing this out. We enlarged some figures (for example figure 3A) to make sure they will be easy to read. All font sizes are now according the guidelines to authors.

References

Baraille F, Ayari S, Carriere V, Osinski C, Garbin K, Blondeau B, Guillemain G, Serradas P, Rousset M, Lacasa M, Cardot P, Ribeiro A (2015) Glucose Tolerance Is Improved in Mice Invalidated for the Nuclear Receptor HNF-4gamma: A Critical Role for Enteroendocrine Cell Lineage. **Diabetes** 64: 2744-2756

Kishida K, Pearce SC, Yu S, Gao N, Ferraris RP (2017) Nutrient sensing by absorptive and secretory progenies of small intestinal stem cells. **Am J Physiol Gastrointest Liver Physiol** 312: G592-G605

Klein AP, Wolpin BM, Risch HA, Stolzenberg-Solomon RZ, Mocci E, Zhang M, Canzian F, Childs EJ, Hoskins JW, Jermusyk A, Zhong J, Chen F, Albanes D, Andreotti G, Arslan AA, Babic A, Bamlet WR, Beane-Freeman L, Berndt SI, Blackford A et al (2018) Genome-wide meta-analysis identifies five new susceptibility loci for pancreatic cancer. **Nature Communications** 9: 556

Muñoz J, Stange DE, Schepers AG, van de Wetering M, Koo BK, Itzkovitz S, Volckmann R, Kung KS, Koster J, Radulescu S, Myant K, Versteeg R, Sansom OJ, van Es JH, Barker N, van Oudenaarden A, Mohammed S, Heck AJ, Clevers H (2012) The Lgr5 intestinal stem cell signature: robust expression of proposed quiescent '+4' cell markers. **EMBO Journal** 31: 3079-3091

Rodriguez-Colman MJ, Schewe M, Meerlo M, Stigter E, Gerrits J, Pras-Raves M, Sacchetti A, Hornsveld M, Oost KC, Snippert HJ, Verhoeven-Duif N, Fodde R, Burgering BM (2017) Interplay between metabolic identities in the intestinal crypt supports stem cell function. **Nature** 543: 424-427

Shukla S, Cyrta J, Murphy DA, Walczak EG, Ran L, Agrawal P, Xie Y, Chen Y, Wang S, Zhan Y, Li D, Wong EWP, Sboner A, Beltran H, Mosquera JM, Sher J, Cao Z, Wongvipat J, Koche RP, Gopalan A (2017) Aberrant Activation of a Gastrointestinal Transcriptional Circuit in Prostate Cancer Mediates Castration Resistance. **Cancer Cell** 32: 792-806

Wang J, Zhang J, Xu L, Zheng Y, Ling D, Yang Z (2017) Expression of HNF4G and its potential functions in lung cancer. **Oncotarget** 9: 18018-18028

Thank you for sending us your revised manuscript. We acknowledge the addition of follow up analyses, including experimental analyses better supporting the proposed role of Hnf4g in enterocyte differentiation. We are satisfied with the modifications made and we think that the study is now suitable for publication in *Molecular Systems Biology*.

Before we formally accept your study for publication, we would ask you to address a few remaining editorial issues listed below.

Corresponding Author Name: Michiel Vermeulen
 Journal Submitted to: Molecular Systems Biology
 Manuscript Number: MSB-18-8227